**Long-term spatiotemporal variations and expansion of low-oxygen conditions in the Pearl River estuary: A study synthesizing observations during 1976-2017**

Jiatang Hu[1,2,3,*], Zhongren Zhang[1], Bin Wang[4], Jia Huang[5]

[1] School of Environmental Science and Engineering, Sun Yat-sen University, Guangzhou, 510275, China

[2] Guangdong Provincial Key Laboratory of Environmental Pollution Control and Remediation Technology, Guangzhou, 510275, China

[3] Southern Marine Science and Engineering Guangdong Laboratory (Zhuhai), Zhuhai, 519000, China

[4] Department of Oceanography, Dalhousie University, Halifax, Nova Scotia B3H 4R2, Canada

[5] Zhongshan Research Institute of Environmental Protection Science Corporation Limited, Zhongshan, 528400, China

Correspondence to: Jiatang Hu (hujtang@mail.sysu.edu.cn)

**Abstract.** The Pearl River estuary (PRE) frequently experiences low-oxygen conditions in summer, with large extents of low-oxygen events and a long-term deoxygenation trend being reported recently. In this study, we provide a synthesis of the spatiotemporal patterns and incidence of different low-oxygen levels in the PRE based on the in-situ observations collected from 1976 to 2017, and aim to elucidate the underlying mechanisms of low-oxygen conditions and their changes over the past 4 decades. The long-term observations show that the dissolved oxygen (DO) content in

the PRE had significant temporal variability and spatial heterogeneity. Low-oxygen conditions (DO < 4 mg/L) occurred mostly in the bottom waters of 5-30 meters during summer and early autumn, with locations and severity varying substantially among years. Coastal waters from the southwest of Lantau Island to the northeast of Wanshan Islands were identified as the hotspot area prone to subsurface low-oxygen conditions due to the combined effects of comparatively deep topography, proper residence time and stability of the water column, and enhanced oxygen

depletion related to high phytoplankton biomass. In addition, the low-oxygen waters, either directly imported from the upstream reaches or generated locally and further transported with the estuarine circulation, also had considerable impacts on the oxygen levels in the estuary. As for early autumn, marked low-oxygen conditions were present both in the surface and bottom waters. A large area affected by low oxygen (~4,450 km$^2$) was found in September 2006, where the low-oxygen conditions were comparable to the most severe ones observed in summer. It was formed by the inflows

of low-oxygen waters from the upstream reaches and enhanced oxygen depletion driven by an intricate coupling of physical and biogeochemical processes. Our analysis also reveals an apparent expansion of the summertime low-

oxygen conditions at the bottom of the PRE since the years around 2000, coincident with the major environment changes in the Pearl River region. Overall, the PRE seems to be undergoing a transition from a system characterized by episodic, small-scale hypoxic events (DO < 2 mg/L) to a system with seasonal, estuary-wide hypoxic conditions in summer. Although exacerbated eutrophication associated with anthropogenic nutrient inputs was generally considered as the primary cause for the deterioration of low-oxygen conditions in the PRE, the sharp decline in sediment load may play an important role as well via increasing water transparency and thereby supporting higher and broader phytoplankton biomass in the estuary.

**Keywords:** Dissolved oxygen; Hypoxia; Temporal and spatial variability; Long-term deoxygenation trend; Pearl River estuary

## 1. Introduction

Oxygen is fundamental to biogeochemical processes and life in aquatic environments. Its decline can impose significant impacts on the aquatic ecosystems. When dissolved oxygen (DO) level drops below 2 mg/L, hypoxic conditions emerge and could have a series of undesirable biological and ecological consequences, such as causing extensive mortalities of fish and dramatic changes in the biological community structures and sediment biogeochemical cycles, promoting the release of greenhouse gases, and aggravating the ocean acidification (Zhang et al., 2010; Cai et al., 2011; Middelburg and Levin, 2009; Diaz and Rosenberg, 2008). In recent decades, hypoxia (DO < 2 mg/L) has frequently occurred in estuaries and coastal waters, largely ascribed to the influence of human activities in combination with global changes (Breitburg et al., 2018; Rabalais et al., 2010). Large-scale hypoxic zones have been observed in a variety of coastal systems, including the Baltic Sea, the northern Gulf of Mexico, Chesapeake Bay, Long Island Sound, and the Yangtze River estuary, with substantial increases in the spatial extent, intensity, and duration of hypoxia in recent years (Fennel and Testa, 2019 and references therein).

A great number of studies have been carried out to investigate the short- and long-term changes, underlying processes, and controlling factors of coastal hypoxia. It was clearly shown that the generation and development of hypoxia in densely populated and urbanized estuaries and other coastal systems are closely linked to the intensifying eutrophication induced by anthropogenic nutrient inputs (Rabalais et al., 2010; Fennel and Testa, 2019). For example, the northern Gulf of Mexico experiences large-scale, persistent hypoxia in the bottom water every summer (with an area of 15,000 km$^2$ on average) largely due to the high primary production stimulated by excessive nutrient loads from rivers together with strong density stratification (Bianchi et al., 2010; Rabalais et al., 2002). In addition, physical conditions such as winds and upwelling also play a significant role in regulating the intensity and duration of hypoxia (Yu et al., 2015; Feng et al., 2014). Compared with the northern Gulf of Mexico, the summertime hypoxic zone off the

Yangtze River estuary has both common (i.e. the intense stratification and eutrophication-driven primary production are the key mechanisms controlling hypoxia) and unique features. Its formation and evolution are influenced by a more complex interaction of various water masses, including the Changjiang diluted water, Taiwan warm currents, Kuroshio subsurface water, and upwelled water (Zhang et al., 2018; Wei et al., 2017). Overall, the dynamics of hypoxia in estuaries and coastal waters are controlled by a combination of physical, biological, and chemical processes (e.g., photosynthesis, nitrification, water-column microbial respiration, sediment oxygen uptake), but the predominant ones vary by systems. Also, the susceptibility to developing hypoxic conditions is significantly different among coastal systems, depending on their respective pressures from local pollution loads and capacities to resist such pressures under regional physical and biogeochemical regimes.

Located in the south of China, the Pearl River estuary (PRE; Figure 1) is the central area of the Guangdong-Hong Kong-Macao Greater Bay Area, surrounded by several megacities (including Guangzhou, Shenzhen, and Hong Kong). With the rapidly growing population and socioeconomic development, the PRE receives large inputs of nutrients and diverse contaminants, which has led to a wide range of severe problems in the estuary, such as eutrophication, red tides, hypoxia, and decline of fishery resources (Dai et al., 2008; Harrison et al., 2008; X. Li et al., 2020). There has been a great concern on hypoxia in the PRE since the 1980s. Based on historical observations, Lin et al. (2001) reported for the first time the incidence of bottom-water hypoxia in the PRE. In the follow-up field surveys, hypoxic events were also observed in the western shoal of the Lingdingyang Bay, the outer Modaomen and Huangmaohai Bays, and the waters adjacent to Lantau Island and Wanshan Islands (Su et al., 2017; Cui et al., 2019; Shi et al., 2019; Yin et al., 2004; Lu et al., 2018; Qian et al., 2018; Li et al., 2018; Ye et al., 2013). Extensive research has been conducted to explore the distribution and influential factors of hypoxia in the PRE by means of field observations and numerical simulations. Results showed that hypoxia mainly appeared in the bottom waters during summer, and the freshwater-induced stratification and sediment oxygen consumption are the dominant factors leading to its generation (Zhang and Li, 2010; Wang et al., 2017; Yin et al., 2004); however, due to the combined effects of relatively shallow depth, high water turbidity, strong physical processes, and short duration of vertical stratification (eroded by periodic tidal forcing and strong wind events; Luo et al., 2009; Lu et al., 2018), hypoxia was generally confined to a very small scale and occurred in an episodic and intermittent manner (X. Li et al., 2020; Rabouille et al., 2008). Nevertheless, notable low-oxygen (DO < 4 mg/L) events, with spatial extents much larger than previous ones, have been reported in recent works (Shi et al., 2019; Lu et al., 2018; Su et al., 2017; Li et al., 2018). Su et al. (2017) reported that the areal extent of bottom DO < 2 mg/L in July 2014 exceeded 290 km$^2$, and Li et al. (2018) reported that the extent of bottom DO < 4 mg/L in June 2015 was estimated ~1,500 km$^2$. There seems to be a growth of the areas affected by low-oxygen conditions in the PRE. Collectively, the problem of hypoxia has received great attention, but there is still a lack of in-depth investigation on the temporal and spatial variability of DO in the PRE, and the scope, frequency, and intensity of

hypoxia as well as its long-term changes are poorly understood. Besides, most of the previous studies focused on the summertime hypoxia, while it is unclear whether hypoxia also exists in other periods. This is an important issue that has long been ignored. In fact, as we will point it out in this study, there was a large area of hypoxia occurring in early autumn as well, which appears to have different features and driving mechanisms from those typically observed in summer.

Here we synthesize the observational data at sites collected during 1976-2017 to explore the temporal and spatial characteristics of DO and the long-term status of low-oxygen conditions (including their locations, areal extents, and frequencies of occurrence) in the PRE. We also aim to clarify the key factors controlling low-oxygen conditions and their expansions over recent years.

## 2. Materials and methods

### 2.1 Data sources, sample collections and measurements

The data used for analysis comprises multiyear cruise observations in the PRE and its adjacent coastal waters compiled from five datasets (see Table 1 for details). The first dataset (Dataset 1) includes vertical profiles of salinity, temperature, DO, nutrients (ammonia, nitrate, and phosphate), and suspended sediment concentrations (SSC) collected by the South China Sea Environmental Monitoring Center during 1976-2006 (with 42 cruises in total); data on

chlorophyll $a$ (chl $a$) was available in September 2006. Part of the data were also used for analysis by X. Li et al. (2020), in which the methods of sampling and chemical analysis were described in their section 2.1. The sample collection, storage and transportation, seawater analysis, and data processing and quality control were strictly operated in accordance with the specifications of oceanographic survey (e.g., GB/T 12763-1991; GB/T 12763-2007) and the specifications for marine monitoring (e.g., GB 17378-1998; GB 17378-2007) issued by the National Standard of P.R.

China. By following these specifications, three-point samples were collected from the surface (0.5 m below the sea surface), half depth, and bottom (0.5-2 m above the sea bed) when the water depth was > 10 m; two-point samples were collected from the surface and bottom when the depth was between 5 and 10 m; and only surface sample was collected when the depth was < 5 m. Temperature was measured on board using a thermometer, and salinity was measured with an induction salinometer in the laboratory. Ammonia ($NH_4$), nitrate ($NO_3$), and phosphate ($PO_4$) were

analyzed using the indophenols blue spectrophotometric, Cd reduction, and phosphorus molybdenum blue spectrophotometric methods, respectively. SSC was measured by the gravimetric method, and chl $a$ was measured using a spectrophotometer after the acetone extraction. As for DO, water samples were collected in brown frost-mouth bottles, immediately fixed with solutions of $MnCl_2$ and KI on board, and were analyzed using the Winkler titration method (Parson et al., 1984). According to the requirements of data quality control, double-parallel samples were

obtained to ensure the accuracy and comparability of the sample measurements.

The second dataset (Dataset 2) contains historical measurements of water quality parameters collected from a summer cruise carried out by the Pearl River Estuary Pollution Project in July 1999 (Chen et al., 2004). The vertical profiles for temperature, salinity, turbidity, DO, and chl *a* were measured using a YSI-6600 multi-parameter automatic water quality sensor. The instrument was calibrated twice with standard samples. The chl *a* data obtained were compared with those obtained from 169 water samples measured by Turner Designs 10-005R fluorescence method, and the DO content was calibrated against the saturation level prior to each profile measurement (Yin et al., 2004). As for nutrients, samples were collected by Go-flo water samplers from the surface, middle, and bottom, and were measured on board with traditional standard methods following the same specification as for Dataset 1. The physical and biochemical parameters of Dataset 2 have been used in multiple observational studies (e.g., Yin et al., 2004; Yang et al., 2011) and modeling studies (e.g., Hu et al., 2009; Luo et al., 2009).

The third dataset (Dataset 3) includes the observed salinity, temperature, and DO profiles collected by the State Oceanic Administration of China from the cruises in four different seasons during 2006-2007, while the fourth dataset (Dataset 4) contains the water quality data (listed in Table 1) collected by the Marine and Fishery Environmental Monitoring Center of Guangdong Province from four seasonal cruises during 2013-2014. These two datasets both followed the same specifications as for Dataset 1 in terms of sampling procedures and chemical analysis. It is important to note that although the specifications issued by the National Standard of China have changed over time, the methodology and fundamental principles for analyzing salinity, DO, nutrients, and chl *a* involved in this study have not changed, ensuring the accuracy and comparability of the data. The last dataset (Dataset 5) is comprised of recent observations on bottom-water DO obtained from literature sources, including data in July 2014 (Su et al., 2017), July 2015 (Lu et al., 2018), and July 2017 (Shi et al., 2019). All these DO data were measured on board using the classic Winkler titration method (Parson et al., 1984). For further details see descriptions in the corresponding literatures.

In addition, the DO saturation state ($DO_S$) and apparent oxygen utilization (AOU) were used to assess the severity of oxygen deficits in the PRE. The $DO_S$ metric (in unit of %) was calculated as the ratio of the in-situ DO concentration to its saturation level ($DO_{sat}$) at a given salinity and temperature, while the AOU was defined as the difference between $DO_{sat}$ and DO, with negative AOU values indicating super-saturation status:

$$DO_S = DO/DO_{sat} \cdot 100 \tag{1}$$

$$AOU = DO_{sat} - DO \tag{2}$$

The DO saturation concentration (i.e. $DO_{sat}$) was computed via the following equation (Garcia and Gordon, 1992):

$$DO_{sat} = \exp[A_0 + A_1 \cdot T_s + A_2 \cdot T_s^2 + A_3 \cdot T_s^2 + A_3 \cdot T_s^3 + A_4 \cdot T_s^4 + A_5 \cdot T_s^5 + S \cdot (B_0 + B_1 \cdot T_s + B_2 \cdot T_s^2 + B_3 \cdot T_s^3) + C_0 \cdot S^2] \tag{3}$$

where $A_0$, $A_1$, $A_2$, $A_3$, $A_4$, and $A_5$ are constants with values of 5.80818, 3.20684, 4.11890, 4.93845, 1.01567, and 1.41575, respectively; $B_0$, $B_1$, $B_2$, and $B_3$ are constants with values of $-7.01211 \times 10^{-3}$, $-7.2595 \times 10^{-3}$, $-7.93334 \times 10^{-3}$, and

-5.5449×10$^{-3}$, respectively; C$_0$ is a constant equal to -1.32412×10$^{-7}$; $S$ is salinity; $T_s$ is a scaled temperature calculated as:

$$T_s = \ln[(298.15 - T)\cdot(273.15+T)^{-1}]\tag{4}$$

where $T$ is temperature (°C).

## 2.2 Data analysis

The spatiotemporal distributions of DO and its related variables in the PRE were visualized using the MATLAB and EXCEL software. Besides, the areal extents of DO < 2, 3, 4 mg/L (referred to as hypoxia, oxygen deficiency, low oxygen, respectively) were also estimated using MATLAB as follows. Firstly, we divided the sea area of the PRE into

165 a number of grid cells with a spatial resolution of 0.01°, and then used the scattered-data-interpolation method (namely, the 'scatteredInterpolant' function) to interpolate the observed DO data onto the grid cells. Secondly, we calculated the total areas for all the grid cells being hypoxic (with DO < 2 mg/L) to estimate the hypoxic areas. Same procedures were applied to compute the areal extents for oxygen deficiency and low oxygen by using a DO threshold of 3 and 4 mg/L, respectively.

To investigate the key factors controlling the occurrence and development of low-oxygen conditions in the PRE, Pearson correlation analysis between DO and other water quality variables was also performed using MATLAB. The Pearson correlation coefficient ($r$) of two variables ($x$ and $y$) is defined as:

$$r = \frac{\sum_{i=1}^{N}(x_i-\bar{x})(y_i-\bar{y})}{\sqrt{\sum_{i=1}^{N}(x_i-\bar{x})^2 \sum_{i=1}^{N}(y_i-\bar{y})^2}}\tag{5}$$

where $N$ is the total number of observations for the variables; the overbar denotes the mean of each variable.

## 3. Results

### 3.1 Seasonal variations of DO-related variables in the PRE

Figure 2 shows the spatial means and standard deviations of DO concentrations, DO$_S$, AOU, salinity, and temperature at the surface and bottom of the PRE during different seasons (note that the lowest DO values observed in each time period were also shown in the figure). It can be clearly seen that all these variables had significant seasonal variations.

In spring and winter, the average DO concentration was maintained at about 6-9 mg/L (Figure 2a1). The DO level at the bottom was slightly lower than that at the surface. No low-oxygen water with DO < 4 mg/L was found except in May 2014. In spring, the surface salinity primarily varied between 20 and 27, and the bottom salinity was slightly higher (Figure 2a4). In winter, salinity was high, with very small difference between the surface and bottom, indicating that the water column was well mixed; the average water temperature (17-21 °C) was lower than that in spring overall

(Figure 2a5). DO$_S$ in these two seasons were similar, mostly at near-saturation (90%-107%; Figure 2a2), while the AOU primarily varied between 0.76 and -0.59 mg/L correspondingly (Figure 2a3).

In summer, the DO content was lower than those in spring and winter, especially in the bottom water (with averages of 3.8-5.8 mg/L; Figure 2b1). Low oxygen (DO < 4 mg/L) was observed in all 19 summer months except August 1976, and oxygen deficiency (DO < 3 mg/L) was observed in 14 out of 19 summer months including July 1987, July of 1991-1992 and 1994-1997, July-August 1999, July 2005, July-August 2006, August 2009, and August 2013. Hypoxia (DO < 2 mg/L) occurred in July of 1987 and 2005. Actually, the DO levels in July of 1997 and 1999 were very close to hypoxic conditions as well, with the minimum concentrations of 2.1 mg/L. Water temperature in summer was comparatively high, with 27.0-30.5 °C at the surface and mostly 24.5-28.5 °C at the bottom (Figure 2b5). Besides, the differences of temperature and salinity between the surface and bottom were evident, especially for salinity (Figure 2b4), showing the presence of pronounced vertical stratification across the water column. Most of the surface DO$_S$ was maintained at 80%-104% (Figure 2b2), but the bottom DO$_S$ was significantly lower (51%-79%). The AOU showed a similar pattern, with higher values at the bottom (1.33-3.57 mg/L) than those at the surface (-0.59-1.95 mg/L; Figure 2b3).

In autumn, water temperature has decreased compared with summer, but was higher than those in spring and winter, as shown in Figure 2c5. As for salinity, it was close to that in spring. Low-oxygen water was absent in mid-to-late autumn (October and November), when the average DO, DO$_S$, and AOU were generally maintained at 6.0-7.5 mg/L, 74%-101%, and -0.034-1.92 mg/L (Figure 2c1-c3), respectively, with small vertical differences. However, in early autumn (September) fairly low DO values appeared both in the surface and bottom waters (as low as 1.1 mg/L at the surface and 0.8 mg/L at the bottom in September 2006). The bottom DO$_S$ could drop below 35% in September 2006 (Figure 2c2), while the AOU could reach as high as 4.16 mg/L (Figure 2c3). This reveals the existence of potential hypoxic events in periods other than summer.

To further explore the spatial characteristics of DO in the PRE, Figures 3 and 4 present the DO distributions at the surface and bottom in different seasons of three decades. As shown, the surface DO in March was high overall and homogenous in space (Figure 3a1), but in April and May, the spatial difference became relatively large (Figure 3b1 and c1). The surface water with DO < 5 mg/L appeared near the eastern four river outlets (see Figure 1 for their locations) in April 2007 and May 2014 (with DO as low as 3.5 mg/L). With respect to summer, the surface DO at the open sea was higher than that inside the estuary and often had high values, for example, in the coastal water surrounding Wanshan Islands (Figure 3b2 and c2); low oxygen was observed in the inner Lingdingyang Bay. As for autumn, the surface DO showed a northwest-southeast distribution pattern in October (Figure 3a3 and b3), relatively high on the western side of the PRE, and it exhibited a northwest-southeast pattern in November (Figure 3c3).

Compared with other seasons, the surface DO in winter was higher, mostly exceeding 7 mg/L, and it was more uniformly distributed over the PRE (Figure 3b4 and c4).

Regarding the bottom DO, its distribution pattern in March closely resembled the surface one, showing small spatial difference as well (Figure 4a1). Since then, the DO content became progressively lower from April to May (Figure 4b1 and c1), and low oxygen appeared near the eastern river outlets, similar to that at the surface. In summer, the bottom water was frequently subject to apparent low-oxygen conditions. As shown in Figure 4a2, there was a low-oxygen zone extending from the west of Lantau Island to Wanshan Islands in July 1997, and the observed areas of low oxygen (HA$_4$) and oxygen deficiency (HA$_3$) were estimated 704 km$^2$ and 157 km$^2$, respectively. In July-August 2006, a relatively large areal extent of low oxygen was observed, distributed in patchy waters (Figure 4b2); the HA$_4$ in total reached 955 km$^2$. In autumn, the bottom water was re-oxygenated and reinstated to relatively high DO levels (Figure 4a3-c3), displaying spatial patterns analogous to those at the surface. It is also noted that the phenomenon of DO < 5 mg/L was observed in the bottom water adjacent to the eastern river outlets in November 2013 (Figure 4c3) as well as in February 2014 (Figure 4c4).

## 3.2 Long-term changes and hotspot areas of low-oxygen conditions in the summer of the PRE

In order to gain a thorough insight into the interannual and long-term changes of the summertime low-oxygen conditions, Figures 5 and 6 present the DO distributions at the surface and bottom of the PRE in the summer months during 1976-2017 (note that the distributions in 1997, 2006 and 2013 have been shown in Figures 3-4 and will not be repeated here). In the surface water, the average DO content varied between 5.7 mg/L and 8.3 mg/L, and high DO values (> 8.0 mg/L) were often observed at the open sea (Figure 5). It is interesting to note that the lowest DO observed at the surface before 1999 was 4.8 mg/L (Figure 2b1), which is well above the threshold of low oxygen, but since 1999, low-oxygen water with DO < 4 mg/L frequently appeared in the inner Lingdingyang Bay; for example, surface DO levels as low as 3.0 mg/L and 2.8 mg/L were observed in July 2005 (Figure 5c3) and August 2013 (Figure 3c2), respectively, as also shown in Figure 2b1. This emerging exacerbation of low-oxygen conditions in the surface water was primarily attributed to the influence of low-oxygen inflows from the upstream reaches (Zhai et al., 2005; He et al., 2014) and increasing sewage discharge brought by rapid economic development and urbanization in the Pearl River Delta over the past 30 years (X. Li et al., 2020).

At the bottom, apparent low-oxygen conditions prevailed in all the summer months except August 1976 (a period prior to the implementation of China's reform and opening-up policy), and their locations and spatial extents varied substantially among different periods (Figure 6). Overall, a majority of the low-oxygen events congregated at the bottom of 5-30-m isobaths and were present in the vicinity of Humen, at the near-shore of the east of the inner Lingdingyang Bay (including Shenzhen Bay), along the deep navigation channel (extending from the north of

Neilingding Island to the southwest of Lantau Island), in the shallow water (between 5-10-m isobaths) on the western side of the middle Lingdingyang Bay, at the outer Modaomen-Jitimen-Huangmaohai Bays, and in the waters from the south of Lantau Island to Wanshan Islands (Figure 7a1-a2). On top of this, the coastal area in the lower estuary, extending from the southwest of Lantau Island to the northeast of Wanshan Islands, was the hotspot with high incidence of bottom low-oxygen events, followed by the shallow area east of Hengqin Island (Figure 7b1-b2).

In terms of the spatial extents, low-oxygen conditions in the 1980s and 1990s were restricted within a very small scale, with all the $HA_4$ and $HA_3$ estimated smaller than 760 $km^2$ and 160 $km^2$, respectively (Figure 6). The areal extents of low-oxygen conditions were relatively large in July of 1987 (Figure 6a3), 1994-1995 (Figure 6c2-c3) and 1997 (Figure 4a2). However, since the 2000s, the area affected by low-oxygen conditions has increased, with the $HA_4 >$ 1,600 $km^2$ being frequently reported. For instance, a more fully developed low-oxygen zone was generated in July 1999 (Figure 6d1). Its central area covered the entire waters of the middle part of the PRE. The $HA_4$ and $HA_3$ were estimated 1,602 $km^2$ and 263 $km^2$, respectively. As for July 2005, the $HA_4$ and $HA_3$ reached 2,683 $km^2$ and 304 $km^2$, respectively (Figure 6d3). A majority of the bottom water resided in low-oxygen conditions, and hypoxia (~6 $km^2$) occurred at the near-shore of the inner Lingdingyang Bay. In particular, recent field investigations during 2014-2017 further indicate the occurrence of large-scale low-oxygen events in the PRE (Figure 6e1-e4). The $HA_2$ observed in the bottom waters surrounding Lantau Island and Wanshan Islands on July 13-16, 2014 (Leg 1) reached 338 $km^2$, with the minimum DO of 0.9 mg/L, and the $HA_4$ and $HA_3$ were estimated 1,290 $km^2$ and 1,037 $km^2$, respectively. Subsequently, a hypoxic zone in similar size was found on July 19-27 (Leg 2) as well, primarily located in the offshore water south of Wanshan Islands where DO dropped to 0.2 mg/L. The $HA_4$ and $HA_3$ reached 3,101 $km^2$ and 1,123 $km^2$, respectively. In July 2015, the low-oxygen zone observed was even larger (reaching 4,453 $km^2$), but the hypoxic area was relatively small (~51 $km^2$; Figure 6e3). With respect to July 2017, a large area affected by low oxygen was observed, with the $HA_4$, $HA_3$, and $HA_2$ of 2,863 $km^2$, 962 $km^2$, and 138 $km^2$, respectively (Figure 6e4). Hypoxia was generated mainly in the south of Lantau Island, with the minimum DO of 0.5 mg/L.

### 3.3 Spatial patterns of DO and low-oxygen conditions in the early autumn of the PRE

In addition to the summer season, low-oxygen conditions also existed in the early autumn in the PRE. Figure 8 shows the DO distributions in September of 2001 and 2006. In general, DO displayed remarkable spatial variations in both periods, but their distribution patterns as well as the location and severity of low-oxygen conditions were quite different. In September 2001, DO exhibited a northwest-southeast pattern at the surface (Figure 8a1) and a north-south pattern at the bottom (Figure 8b1). The DO content was relatively low on the eastern side of the PRE. The $HA_4$ in the surface water was estimated 345 $km^2$, while the $HA_4$ and $HA_3$ at the bottom were estimated 1,056 $km^2$ and 270 $km^2$, respectively.

In comparison, spatial variation of the surface DO in September 2006 was more significant, generally showing a northeast-southwest pattern with low DO at the near-shore and high DO at the open sea (Figure 8a2). Low-oxygen water almost covered the entire surface of the inner Lingdingyang Bay, and hypoxia (with DO as low as 1.1 mg/L) occurred in the upper region, with the $HA_2$ reaching 108 km$^2$; the $HA_4$ and $HA_3$ were estimated 1,131 km$^2$ and 512 km$^2$, respectively. In regard to the bottom, most of the PRE was occupied by water low in oxygen, which extended from the inner Lingdingyang Bay further to the western PRE and the southwest of Lantau Island (Figure 8b2). The estimated $HA_2$ reached 333 km$^2$, with the minimum DO of 0.8 mg/L, while the $HA_4$ and $HA_3$ reached 4,448 km$^2$ and 2,061 km$^2$, respectively.

## 4. Discussion

### 4.1 Mechanisms controlling the summertime low-oxygen conditions in the PRE

Our results demonstrate that the PRE has frequently experienced low-oxygen events in summer. For all the 22 summer months investigated (during 1976-2017), there were 21 records with low oxygen, 17 (accounting for 77%) with oxygen deficiency, and 5 (23%) with hypoxia, reflecting a feature of seasonal low oxygen and episodic hypoxia in the PRE. In addition to our study, there also have been many other reports on the low-oxygen and hypoxic events during the summer months, including June of 2001-2005, 2009-2010 and 2015; July of 2000, 2003-2004 and 2010; and August of 2001-2002, 2005, 2007, 2010-2011 and 2017 (Zhai et al., 2005; Yang et al., 2011; Huang et al., 2012; Ye et al., 2013; Li et al., 2018; Qian et al., 2018; X. Li et al., 2020; Cui et al., 2019).

For the PRE, the formation and development of low-oxygen conditions is closely related to multiple factors including the Pearl River diluted water and its carrying terrestrial substances, local productivity, vertical stratification intensity, water residence time, and topography. Specifically, the increased water temperature in summer (Figure 2) could result in notable decrease of oxygen solubility in water and accelerated oxygen consumption by microbial respiration (Breitburg et al., 2018). In addition, the runoff was also increasing, with the summertime discharge approaching 16,500 m$^3$/s (Figure 9). Massive freshwater inputs formed a distinct plume structure at the surface (Figure 10a1) together with the intrusion of high-salinity bottom waters along the deep navigation channel (Figure 10a2). This two-layer circulation driven by the density gradient (Wong et al., 2003) largely determines the distributions of biochemical components and the subsequent development of low-oxygen conditions in the PRE. As shown in Table 2, the bottom DO in summer had a significantly negative correlation with the vertical density difference ($r = -0.7131$, $p < 0.01$). This confirms that intense stratification was a critical physical setting for the generation and maintenance of bottom low-oxygen conditions, as denoted by previous studies (e.g., Rabouille et al., 2008; X. Li et al., 2020).

In the stratified waters (Figure 10a3), the bottom $DO_S$ was comparatively low (Figure 10a4), implying the presence of significant oxygen consumption in the region. Multiple reports have manifested that sediment oxygen

uptake was the dominant oxygen sink in the PRE and fueled primarily by the remineralization of organic matter accumulated in the sediments (Zhang and Li, 2010; Wang et al., 2017), which originated from terrestrial inputs and/or marine-sourced inputs through local production (Su et al., 2017; Ye et al., 2017). The dominance of terrestrial- and marine-sourced organic matter varied greatly in space. More specifically, a large fraction of terrestrial particulate matter settled on the sediments after entering the PRE (Hu et al., 2011) and ultimately experienced diagenetic decomposition by heterotrophic bacteria (Li et al., 2018), resulting in high oxygen demand (Wang et al., 2017; Lu et al., 2018). This was considered the primary cause for the low oxygen and sporadic hypoxic events observed in the shallow water east of Hengqin Island, the outer Modaomen Bay, and the deep channel (Figure 7b1-b2), where considerable deposition of organic matter and intense stratification occurred simultaneously (Hu et al., 2011). While in the offshore area of the PRE, water transparency improved due to the consecutive settling and dilution of suspended particles (Figure 10a5). Data showed that the surface SSC in the offshore waters was significantly lower than that at the nearshore (Figure S1 in the supplementary materials). On top of this, the local flow convergence induced by the cyclonic vortices in this coastal transition zone, which was formed by the interaction between freshwater buoyancy discharge and wind forcing, was also conducive to the long residence time and nutrients accumulation and thus favored the growth of phytoplankton biomass within the region (D. Li et al., 2020). As shown in Figure 10a6, the chl *a* content was higher at the offshore than in the main estuary, especially on the eastern side where algae blooms and red tides have been frequently reported (Dai et al., 2008; Harrison et al., 2008) accompanied by supersaturated DO at the surface (Figures 3 and 5). Moreover, the offshore water of the eastern PRE is comparatively deep (> 15 m; Figure 1) and thus possessed a more stable profile of vertical stratification. The resulting restricted oxygen supply, coupled with the substantial delivery of labile organic matter from the surface water with phytoplankton blooms, made the coastal area from the southwest of Lantau Island to the northeast of Wanshan Islands vulnerable to frequent occurrence of low oxygen and episodic hypoxia (Figure 7b1-b2). The close linkage between the development of bottom low-oxygen conditions and nutrient-stimulated high productivity at the surface for this hotspot (hypoxia-prone) area has also been illustrated in several works (Su et al., 2017; Qian et al., 2018; X. Li et al., 2020), showing the dominant role of marine-sourced organic matter over the terrestrial inputs with respect to oxygen depletion. Moreover, under the control of the two-layer estuarine circulation, the local low-oxygen water generated in the lower estuary could be transported to the inner Lingdingyang Bay along the bottom of the deep channel (Cui et al., 2019; X. Li et al., 2020), which would further exacerbate the low-oxygen conditions in the bay.

## 4.2 Influential factors and underlying mechanisms of the low-oxygen conditions in the early autumn

In contrast to summer, there were very few studies concerning the problem of low-oxygen conditions in the early autumn of the PRE. Although Qian et al. (2018) discovered the existence of low oxygen in the waters adjacent to

Humen as well as Hengqin Island and Lantau Island in September 2010, this issue did not attract much attention because the actual coverage and severity of low oxygen have not been fully unveiled owing to the small-scale field survey. Here the observations we used show prominent low-oxygen conditions in the early autumn (September) of 2001 and 2006 (both with the areal extent of low oxygen $> 1,000$ km$^2$; Figure 8). In particular, over 100 km$^2$ and 330 km$^2$ of hypoxia were observed at the surface and bottom, respectively, in September 2006, which witnessed a hypoxic scale comparable to that of July 2014 (the period with the largest hypoxic area), a low-oxygen area very close to that of July 2015 (the period with the largest low-oxygen area), and an oxygen-deficiency extent that was significantly larger than those in all the summer months. Besides, it should be noted that the low-oxygen conditions in September 2006 were much more severe than those in the summer of 2006 (Figure 4b2), indicating that this issue was not a simple succession of the summer ones, but likely a spontaneous phenomenon that formed independently.

Unlike the summer, the freshwater discharge in September has decreased to approximately 10,000 m$^3$/s (Figure 9), equivalent to ~61% of the summertime discharge, but it was still much larger than those in the mid-to-late autumn and winter. The drawdown of freshwater inputs led to weaker extension of the diluted water (Figure 10b1-b2) and smaller vertical density difference (Figure 10b3) compared with summer. The estuary was weakly stratified overall except for the waters from the outer Modaomen Bay to Wanshan Islands where moderate stratification was found. Table 2 shows that there was no significant correlation between the bottom DO and the vertical density difference in either September 2001 or September 2006. This implies that the mild stratification had a small effect on the low-oxygen conditions in early autumn, which seems to have a different underlying mechanism from that in summer.

As for September 2006, we speculate that the severe hypoxia occurring at the surface and bottom of the inner Lingdingyang Bay (Figure 8a2 and b2) resulted mainly from the inflows of low-oxygen waters from the upstream reaches, as evidenced by the significantly positive correlations between salinity and DO in Table 2 (with the correlation coefficients $r$ reaching 0.7341 and 0.6637 at the surface and bottom, respectively, $p < 0.01$). Other factors including sewage effluents discharged from the runoff and coastal cities (as evidenced by the significant correlation between the surface DO and NH$_4$ concentrations) and the respiration related to phytoplankton imported from the upstream reaches (as indicated by the high regional chl $a$ contents in Figure 10b6; also in accordance with the finding by Ye et al. (2013)) supplement the maintenance of hypoxia. With respect to the waters from the middle part of the Lingdingyang Bay to the Modaomen Bay, marked low-oxygen conditions were largely restricted to the bottom (with DO$_S$ ranging between 40% and 50%; Figure 10b4) and possibly controlled by a complex interplay of various processes. Since the river discharge has weakened compared with summer, a greater fraction of terrestrial organic matter would shift towards to deposit at this area, while the water residence time became relatively longer (Sun et al., 2014) and thus favored the retention of organic matter and its eventual decomposition at the bottom. The influx of low-oxygen waters from the inner Lingdingyang Bay was an additional stressor that could also contribute to the substantial oxygen deficit

in this region. Regarding the lower estuary, a large area of high phytoplankton biomass (with the maximum chl $a$ content $> 23$ mg/m$^3$) was observed (Figure 10b6) because of the good light condition attributed to the low concentrations of suspended particles (Figure 10b5). Meanwhile, the prolonged water residence time was conducive to the retention and degradation of organic matter supplied by the elevated primary production in the region, thus resulting in significant oxygen consumption and decline in the bottom water.

Overall, the combined actions of the upstream low-oxygen inflows and enhanced oxygen depletion driven by an intricate coupling of physical and biogeochemical processes eventually led to the emergency of large-scale (estuary-wide) low-oxygen event in September 2006. By contrast, the low-oxygen conditions were much less severe in September 2001 (Figure 8a1 and b1). This suggests that in the periods of early autumn, there was considerable interannual variability in the spatial extents and intensity of low-oxygen conditions, which has also been observed in summer. Plausible drivers for such interannual variability include the associated changes in freshwater discharge and wind forcing (both of which are the major factors controlling the spreading of the nutrient-rich river plume; Xu et al., 2019) as well as terrestrial material inputs (including organic matter, nutrients, and suspended particles), which could induce significant alterations to physical conditions (e.g., vertical density stratification and mixing, estuarine circulation, retention time) and internal production of organic matter. Further studies are needed to clarify the role of changes related to different dynamic factors in the generation and variations of low-oxygen conditions in the PRE.

### 4.3 Long-term expansions of low-oxygen and hypoxic areas in the PRE

The global ocean and coastal waters have been experiencing notable oxygen declines over the past several decades, with significant increases in the number and severity of hypoxic areas in recent years (Breitburg et al., 2018; Diaz and Rosenberg, 2008). This rapid oxygen loss is largely attributed to intensive anthropogenic activities that have caused global warming and nutrient enrichment of coastal waters. Apparent long-term expansions of hypoxic conditions have been documented in several coastal systems where sustained seasonal hypoxia has been reported. For instance, the hypoxic volume in the Baltic Sea has expanded dramatically with increasing nutrient inputs from its watershed and enhanced water-column respiration resulting from warming (Fennel and Testa, 2019 and references therein). In Chesapeake Bay, the hypoxia can be tracked back to the 1930s and has witnessed an expansion of its volume since the 1950s due to the increased nutrient loads (Hagy et al., 2004). Moreover, the hypoxia in the northern Gulf of Mexico has been documented since 1985 (Rabalais et al., 2002). However, models suggest that the occurrence of large-scale hypoxia can be as early as the 1970s (Scavia and Donnelly, 2007). Despite large interannual variability, the hypoxic area has increased from an average of 8,300 km$^2$ in 1985-1992 to 16,000 km$^2$ in 1993-2001 (Scavia et al., 2003). To mitigate the hypoxia, nutrient reduction plans have been proposed. In addition to the nutrient loads, the long-term climate change can also exaggerate hypoxia and reduce the positive impacts from nutrient reduction. Modeling studies

have suggested that the worsened physical conditions since the 1980s in Chesapeake Bay, e.g. prolonged vertical exchange time and elevated temperature, can contribute to the increased hypoxia (Du and Shen, 2015; Du et al., 2018).

A more recent study shows that the impacts from climate change and nutrient reduction cancel out and therefore the hypoxic volume in Chesapeake Bay shows no significant long-term trends in the past three decades (Ni et al., 2020). Similar findings have been also archived in other hypoxic systems, e.g. the northern Gulf of Mexico (Kemp et al., 2009; Obenour et al., 2013). However, it has to be noticed that the susceptibility of hypoxia to increased anthropogenic activities varies across different coastal systems due to their physical and biological features.

It is commonly recognized that the PRE did not develop similar large-scale, persistent low-oxygen zone as in other hypoxic systems (e.g., the northern Gulf of Mexico, Chesapeake Bay). A combination of intriguing features including shallow and turbid waters, rapid physical exchanges, and unstable vertical stratification provides good buffering capacity for the PRE to mitigate eutrophication and hypoxic conditions in summer. Moreover, the freshwater input was characterized with excess nitrogen (N) and low phosphorus (P), yielding an N: P ratio beyond 100 (Harrison
et al., 2008; Hu and Li, 2009). Therefore, the growth of phytoplankton was heavily inhibited over a large area of the PRE due to strong phosphorus limitation, together with high water turbidity (Figure 10a5 and Figure S1) and short residence time (Sun et al., 2014). However, with the rapidly changing environments in recent years, the summertime low-oxygen conditions in the PRE has undergone an apparent expansion in areal extents, as indicated in Figures 4, 6 and 11a1; large extents of low oxygen, oxygen deficiency, and hypoxia exceeding 4,450 $km^2$, 1,120 $km^2$, and 330 $km^2$,
respectively, were present during 2014-2017. This emerging declining trend of bottom oxygen and spatial expansion of low-oxygen conditions in the PRE have been supported by previous studies (Ye et al., 2012; Qian et al., 2018). By analyzing the sediment cores in the PRE, Ye et al. (2012) discovered that due to the influence of human activities, eutrophic conditions in the estuary have been exacerbated since the 1970s, and the relative abundance of hypoxia-resistant foraminifera has substantially increased, implying that the DO conditions in the bottom waters were
deteriorating. Based on the monitoring data collected by the Hong Kong Environmental Protection Agency from a coastal site (SM18) during 1990-2014, Qian et al. (2018) also identified a significant decrease in the bottom DO over the past 25 years. Collectively, the results from our study and previous research described above have consistently corroborated the gradual exacerbation of low-oxygen conditions in the PRE. Besides, the years around 2000 appear to be a key time node for the emergence of apparent deoxygenation, which coincided with the sharp environment changes
along with the growing economics and populations in China.

Firstly, the long-term decline of oxygen was thought to be principally driven by the increasing wastewater discharge in the Pearl River Delta region. As shown in Figure 11b1, the annual wastewater discharge in Guangdong Province has increased from ~2.5 billion tons in 1990 to ~9.4 billion tons in 2016, thereby resulting in remarkable nutrient enrichment and water quality deterioration. Figure 11a2-a3 showed a long-term increasing trend in the nutrient

concentrations near the eastern four outlets; although there existed data gaps in certain years, it is still clear that the nutrient concentrations after 2000 were higher than those before. According to X. Li et al. (2020), nutrient concentrations in the upstream reaches, which received a large amount of sewage effluents from Guangzhou and Dongguan, were mostly higher than 50 μg/L for $NH_4$, 1000 μg/L for $NO_3$, and 30 μg/L for $PO_4$ since 2000, coincident with the accelerated growth of wastewater discharge (Figure 11b1). The elevated nutrients, especially $PO_4$, which almost doubled from the 1990s to the 2010s (X. Li et al., 2020; Figure 11a3), would very likely promote the phytoplankton biomass and local production of organic matter in the PRE; indeed, X. Li et al. (2020) reported an increasing trend of chl *a* in the surface waters of the lower estuary from 2001 to 2011, with concentrations exceeding 10 μg/L in recent years. Furthermore, as a result of intense nitrification and aerobic respiration of organic matter from direct anthropogenic inputs (He et al., 2014), the DO content in the upstream reaches has been declining since 1998, with periodic low oxygen and episodic hypoxia starting from 2000 (X. Li et al., 2020). Therefore, the inflows from the upstream reaches preconditioned the freshwater delivering into the PRE with increasingly high levels of nutrients and organic matter as well as low-oxygen waters, which had significant contributions to the low DO levels in the inner Lingdingyang Bay since 1999 (Figures 5-6).

Secondly, in addition to the continued rise in nutrients and various pollutants, dramatic alterations to sediment load were identified as well because of the regional high-intensity human activities (e.g., the construction of dams and reservoirs, soil-water conservation measures, altered land-use patterns). As recently reported by Wu et al. (2020), the sediment load of the nine major rivers (including the Yangtze, Pearl, and Yellow rivers) in China has dramatically dropped by 85% over the past six decades, and the year 1999 was identified as one of the important time nodes for the sediment declines. As for the Pearl River, its water discharge showed a slight downward trend during 1979-2015 (Figure 11b2); however, the sediment load was characterized with a more significant decline (Figure 11b3), approximately by 63% between 1979-1998 (~85 million tons/a) and 1999-2015 (~31 million tons/a) primarily attributed to anthropogenic impacts (Wu et al., 2020). The observed data also showed that the averaged values of SSC in the PRE in the 2000s-2010s were generally lower than those in the early 1990s (Figure S1), which is consistent with the long-term declining trend of the sediment load. One potential consequence of this abrupt change was that the PRE would become more susceptible to widespread eutrophication and strengthened oxygen depletion as the light shading effect of suspended sediments on primary production was greatly weaken. This inference could be supported to a certain extent by the negative correlation between DO and SSC (i.e. the lower SSC, the higher DO) in the surface waters of the PRE  (Figure S2), which suggests that with the decrease of SSC, water transparency would be greatly improved and conducive to the growth of phytoplankton; therefore, the surface DO level increased as a result of the oxygen release via photosynthesis, whereas the bottom DO consumption would be enhanced due to the substantial supply and decomposition of organic matter associated with the aggravated eutrophication. With respect to other

stressors, including ocean warming and upwelling of subsurface low-oxygen, nutrient-rich waters from the South China Sea, it has been clarified that they had minor contributions to the observed long-term decline of oxygen in the PRE (Qian et al., 2018).

Collectively, the significant decline of the Pearl River sediment load, superimposed with the considerable changes in nutrients, would play an important role in controlling the expansion of low-oxygen conditions in the PRE. Future work is needed to further clarify the role and relative contributions of these changes in the long-term deoxygenation trend in the PRE.

### 4.4 Implication and limitations

By using the field observations over 42 years, this study is a first attempt to reveal the long-term evolution of low-oxygen conditions in the PRE in terms of spatial extents. Although there existed data gaps in certain years and lack of conformity in observational coverage, the observations witnessed a distinct exacerbation of summertime low-oxygen conditions as the increased frequencies in extremes (e.g. $HA_4 > 1,600$ km$^2$; Figure 11a1). In the context of global oxygen declines, unveiling this potential decadal change in low-oxygen conditions could be helpful for us to project

and better understand the future oxygen status in the PRE as well as other coastal systems subject to intense anthropogenic disturbances. In addition, our results also identified two prominent low-oxygen events in the early autumn of the PRE, which were not inherited from summer ones but formed by different mechanisms. Given the insufficient attention to this issue so far, our finding is not merely a supplement to the understanding of oxygen dynamics in the PRE, but also a critical reminder for the community to realize the importance and severity of the low-

oxygen problem in early autumn. It is highly essential to strengthen scientific research and field investigations on this issue in order to fully elucidate its current status, formation process, and controlling factors, especially in the context of the spatial expansions of low-oxygen conditions observed in summer.

Nevertheless, a caveat to the historical observations for the PRE is that they were under sampled in some years, especially before the 2010s. Besides, the available amounts of different data types were also different; the historical

data on DO, nutrient concentrations, temperature, and salinity were relatively abundant, while the long-term data on chl *a* and nutrient loadings were lacking. This would largely limit our ability to quantify the long-term changes in low-oxygen conditions and to further investigate the associated mechanisms. For instance, low oxygen was observed in some summer months (Figure 6), e.g. July 1992, July 1994, and August 1999; however, we were unable to estimate their coverage and intensity due to the lack of sufficient observations. It is thus important to note that cautions should

be kept when interpreting these low-oxygen areal extents estimated from the available data (Figure 11a1), which is inappropriate to be directly used for quantifying the long-term oxygen changes due to the data limitations. To this end, we would like to emphasize the importance of conducting estuary-wide surveys to collect extensive data on DO and its

related factors in the PRE in a consistent way. Furthermore, we could also collect more in-situ observations to fill the data gaps and merge the data into sophisticated numerical models through model calibration and/or data assimilation to provide a more thorough insight into the temporal and spatial variability, development, and underlying processes of low-oxygen conditions in the PRE.

## 5. Conclusion

This study explores the long-term spatiotemporal variations of DO as well as the locations and severity of low-oxygen conditions in the PRE by utilizing a collection of observations during 1976-2017. Our analysis has revealed a number of important aspects concerning the low-oxygen status (including the hotspots with high incidence of low-oxygen events) and the associated long-term changes over the past 42 years. The spatial patterns of DO and low-oxygen conditions exhibited significant seasonal, intra-seasonal, and interannual variations. Low-oxygen conditions were frequently observed in summer and primarily present in the bottom waters affected by intense vertical stratification and oxygen uptake by the sediments. Furthermore, the summertime low-oxygen conditions have experienced an apparent expansion in spatial extents over recent years. The synergetic effects of substantially increased loads of anthropogenic nutrients and organic matter, sharply decreased load of suspended sediments, and direct inflows of low-oxygen waters from the Pearl River act to promote the exacerbation of low-oxygen conditions in the PRE. Prominent low-oxygen events were also present in early autumn, showing different characteristics and underlying mechanisms from those in summer. To sum up, our results indicate that the PRE could form large areas of low oxygen under proper environmental conditions, as exemplified in September 2006 and the summer months of 2014-2017, and that this river-dominated estuary has shown a clear trend of developing into a seasonal, estuary-wide oxygen-deficiency/hypoxic zone in summer.

*Data availability.* The in-situ observations in July 1999 and 2013-2014 will be available at a public data storage, while the oxygen data in July of 2014-2017 derived from literatures can be downloaded directly via the links provided in the corresponding literatures.

*Author contributions.* **Jiatang Hu**: Conceptualization, data analysis, drafting, review & editing. **Zhongren Zhang**: Graphic visualization, review & editing. **Bin Wang**: Writing & review. **Jia Huang**: Data compilation & analysis.

*Declaration of competing interests.* The authors declare that they have no known competing financial interests or personal relationships that could have appeared to influence the work reported in this paper.

*Acknowledges.* This work was supported by the Joint Research Fund of the National Natural Science Foundation of China and Guangdong Province (U1901209).

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

## Tables

**Table 1. A summary of the observational data in the PRE used for analysis.**

| Dataset | Time period | Data in use |
|---|---|---|
| **(1)** | spring (March of 1985, 1987, and 1989-1997; April 2006) summer (August 1976; July of 1985, 1987, and 1989-1997; August 1999; July 2005; August 2009) autumn (November 1976; October of 1985, 1987, and 1989-1997; September of 2001 and 2006) Winter (February 1978) | Temperature (T), salinity (S), dissolved oxygen (DO), suspended sediment concentrations (SSC), ammonia ($NH_4$), nitrate ($NO_3$), phosphate ($PO_4$), and chlorophyll (chl) $a$* in the surface and bottom waters |
| **(2)** | summer (July 1999) | Surface and bottom T, S, DO, SSC, $NH_4$, $NO_3$, $PO_4$, and chl $a$ |
| **(3)** | spring (April 2007); summer (July-August 2006); autumn (October-November 2007); winter (December 2006-January 2007) | Surface and bottom T, S, and DO |
| **(4)** | spring (May 2014); summer (August 2013); autumn (November 2013); winter (February 2014) | Surface and bottom T, S, DO, SSC, $NH_4$, $NO_3$, and $PO_4$ |
| **(5)** | summer (July of 2014, 2015, and 2017) | Bottom DO |

Data sources: (1) 42 cruises conducted by the South China Sea Environmental Monitoring Center during 1976-2006; (2) a summer cruise conducted by the Pearl River Estuary Pollution Project; (3) 4 seasonal cruises conducted by the State Oceanic Administration of China during 2006-2007; (4) 4 seasonal cruises conducted the Marine and Fishery Environmental Monitoring Center of Guangdong Province during 2013-2014; (5) data adopted from the literatures (Su et al., 2017; Lu et al., 2018; Shi et al., 2019).

* Note that in dataset (1) chl $a$ data was only available for September 2006.

**Table 2. Pearson correlation coefficients ($r$) between DO and other water quality metrics**

| Time period | | S | T | $\triangle\rho$ | NH$_4$ | NO$_3$ | PO$_4$ | chl $a$ |
|---|---|---|---|---|---|---|---|---|
| **Summer** | Surface | 0.2082[**] | -0.0043 | 0.2820[**] | -0.3188[**] | -0.3939[**] | -0.6153[**] | 0.6788[**] |
| | Bottom | -0.5977[**] | 0.4292[**] | -0.7131[**] | 0.1109 | 0.2926[**] | -0.1421 | 0.4035[*] |
| **Sep 2006** | Surface | 0.7341[**] | -0.3000 | 0.6783[**] | -0.4910[*] | -0.8052[**] | -0.2742 | 0.4314 |
| | Bottom | 0.6637[**] | -0.6703[**] | 0.4582 | 0.0417 | -0.6693[**] | -0.3513 | -0.6077[**] |
| **Sep 2001** | Surface | 0.0394 | -0.0989 | 0.1424 | -0.5686[*] | 0.1554 | -0.5381[*] | NA |
| | Bottom | -0.2953 | 0.3261 | -0.1002 | -0.4612 | 0.3292 | -0.4097 | NA |

Notes: (1) S and T are salinity and temperature, respectively; $\triangle\rho$ represents the difference of density between the surface and bottom waters (bottom density minus surface density); NH$_4$, NO$_3$, and PO$_4$ are the concentrations of ammonia, nitrate, and phosphate, respectively; chl $a$ represents the chlorophyll $a$ content. (2) [**] indicates significance correlations at $p < 0.01$, and [*] indicates significant correlations at $p < 0.05$.

## Figures

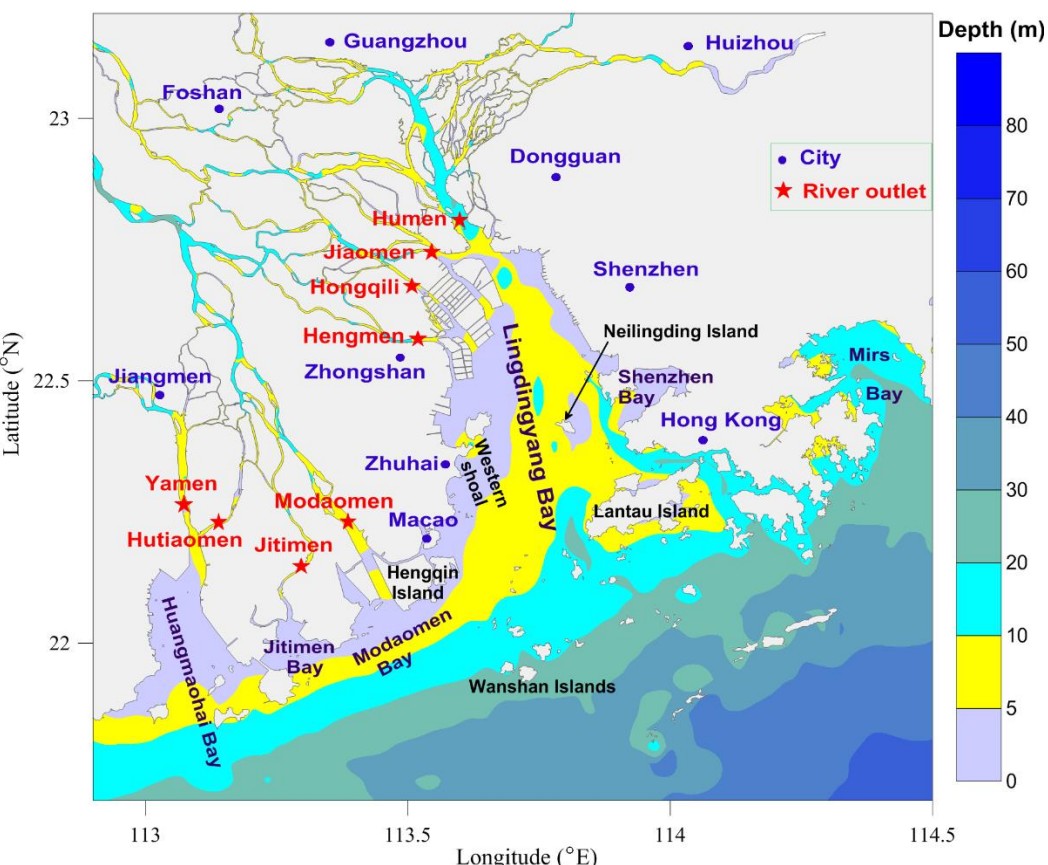

**Figure 1**. Map of the Pearl River estuary (PRE) and adjacent coastal waters. Note that the blue dots denote cities in the Guangdong-Hong Kong-Macao Greater Bay Area, and the red stars indicate the locations of eight outlets of the Pearl River freshwater discharged into the PRE; Humen, Jiaomen, Hongqili, and Hengmen are typically called the eastern four river outlets, while the others are called the western four river outlets.

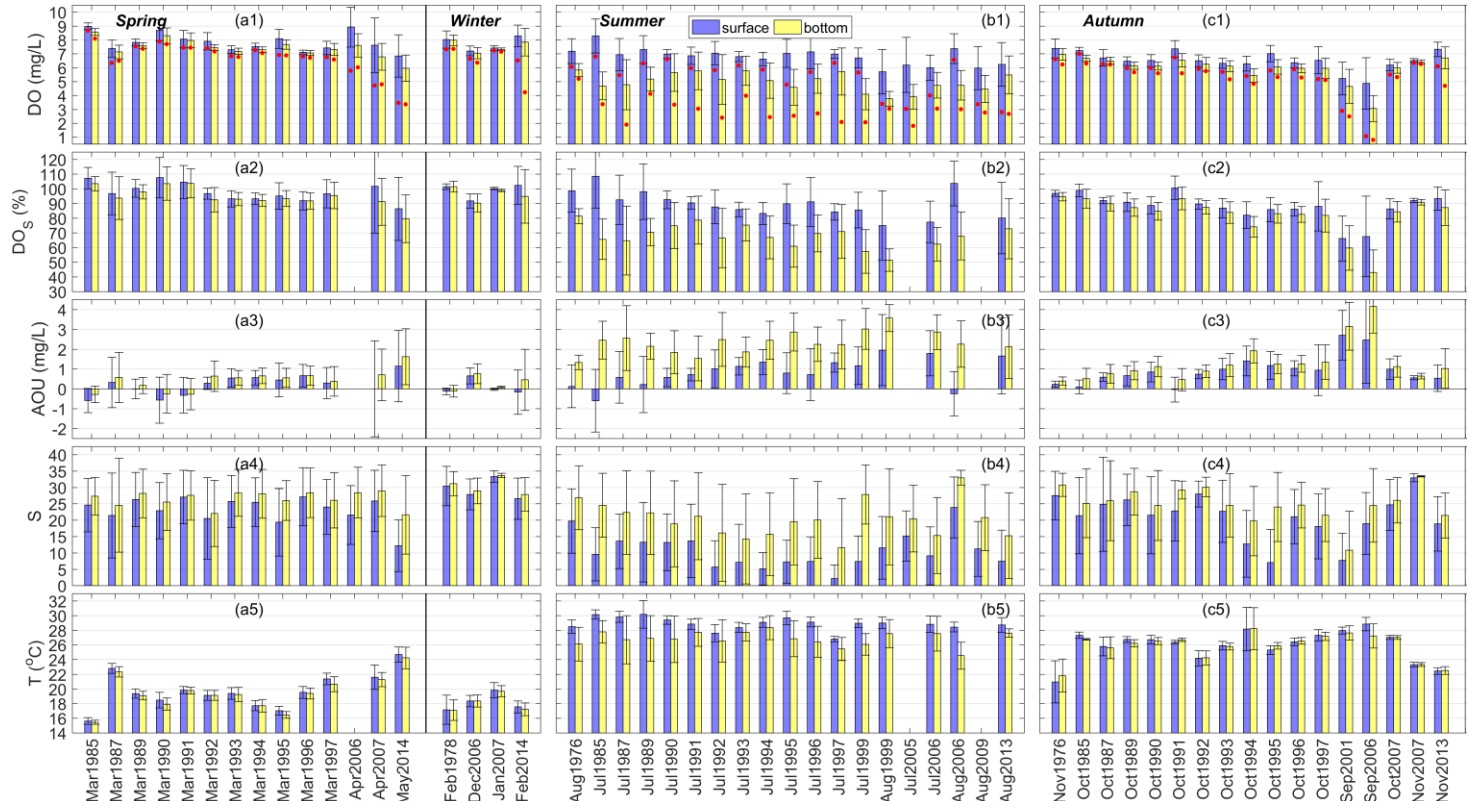

**Figure 2**. Spatial means and standard deviations of DO concentrations, DO saturation (DO$_S$), apparent oxygen utilization (AOU), salinity (S), and temperature (T) in the surface and bottom waters of the PRE in (a) spring (March-May) and winter (December-February), (b) summer (June-August), and (c) autumn (September-November) during 1976-2014. Note that the red dots in the first row of the figure represent the lowest DO values measured in each time period.


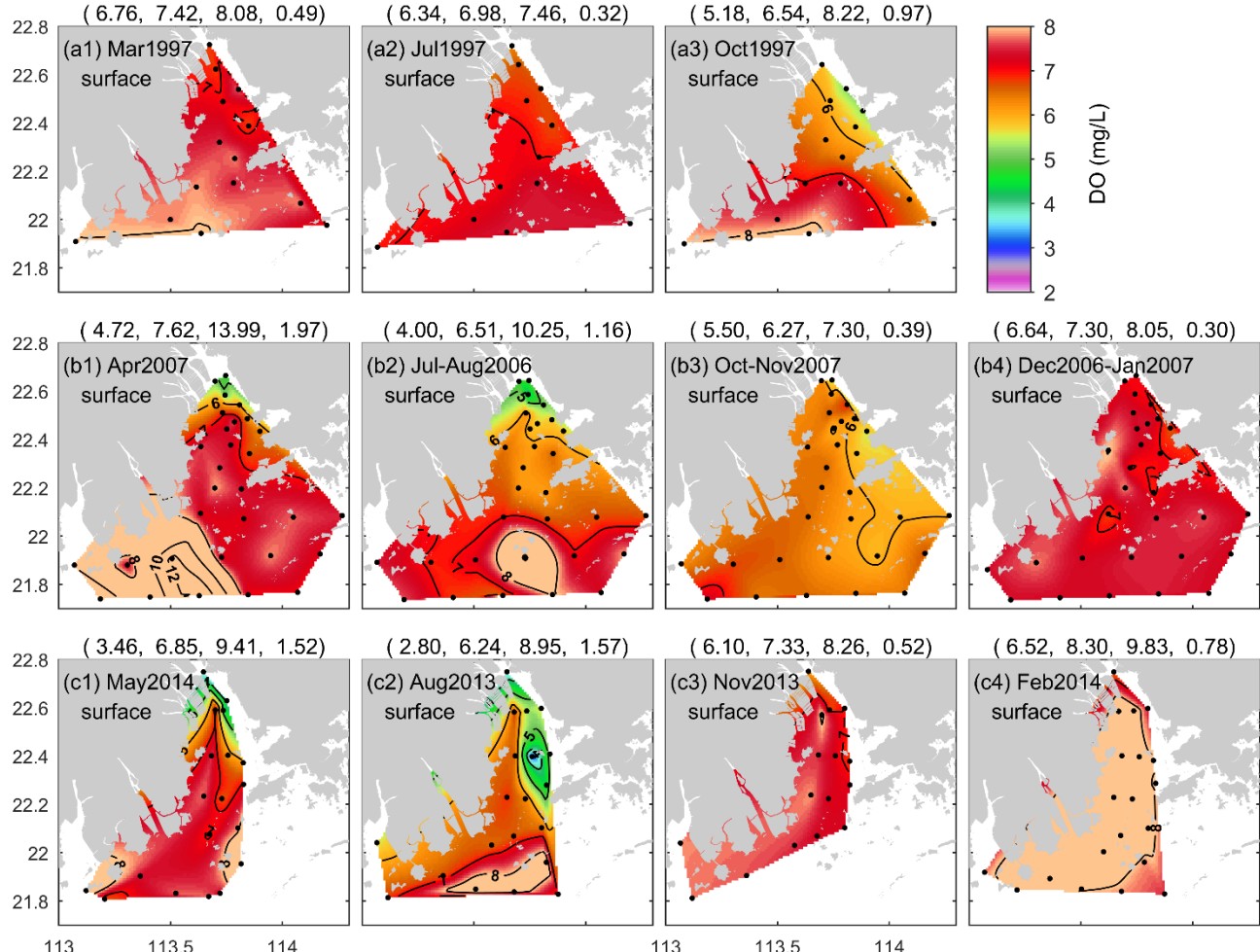

**Figure 3**. Seasonal variations of surface DO distributions in the PRE for (a) 1997, (b) 2006-2007, and (c) 2013-2014. The first to last columns of the figure correspond to spring, summer, autumn, and winter, respectively. Note that the numbers in brackets in the titles are the minimum, mean, maximum, and standard deviation values of DO in sequence; 725    the black dots show the locations of the sampling stations.

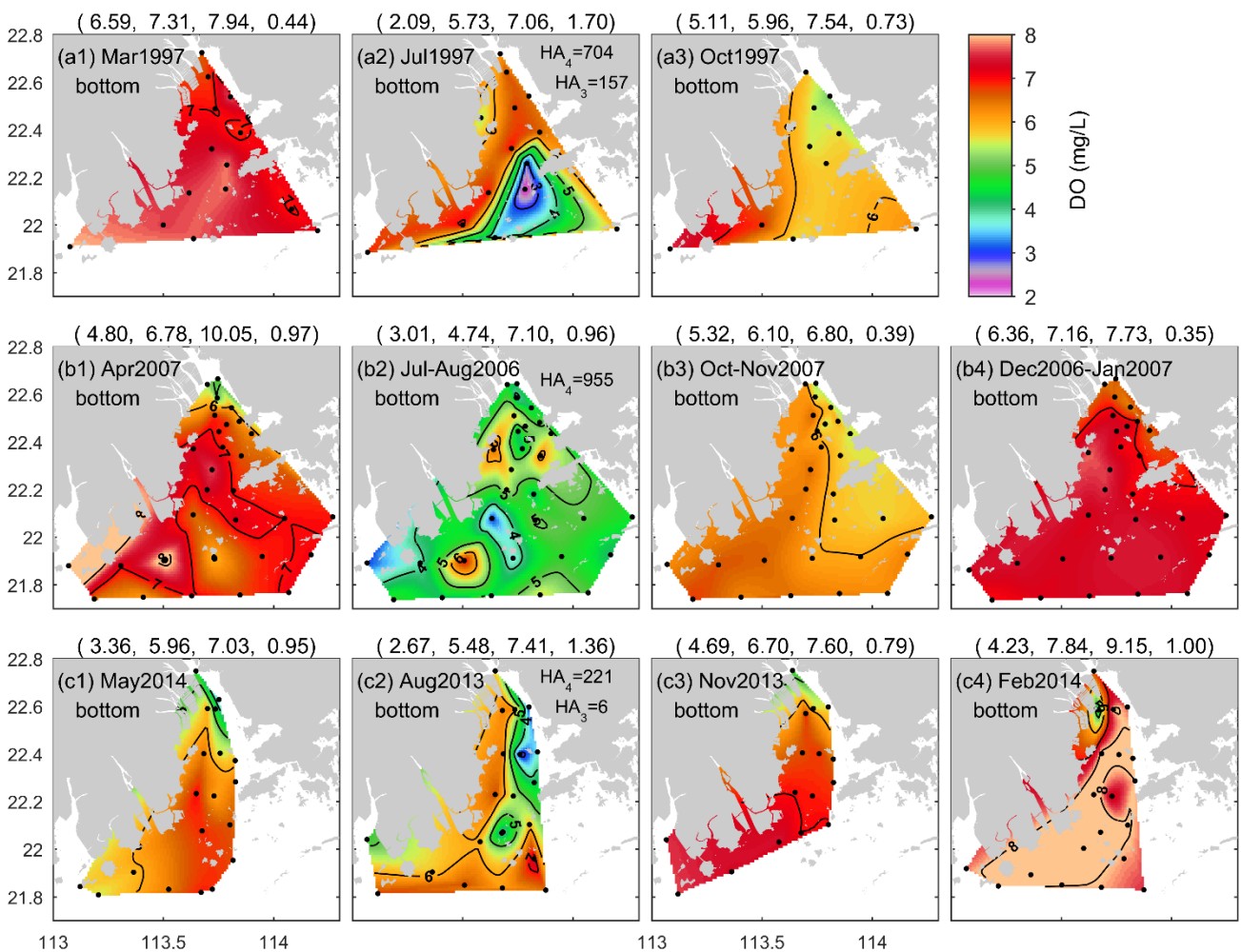

**Figure 4**. Same as Figure 3 but for the bottom water. Note that $HA_4$, $HA_3$, and $HA_2$ represent the areal extents ($km^2$) with DO $<$ 4, 3, and 2 mg/L estimated from the available monitoring data, respectively.

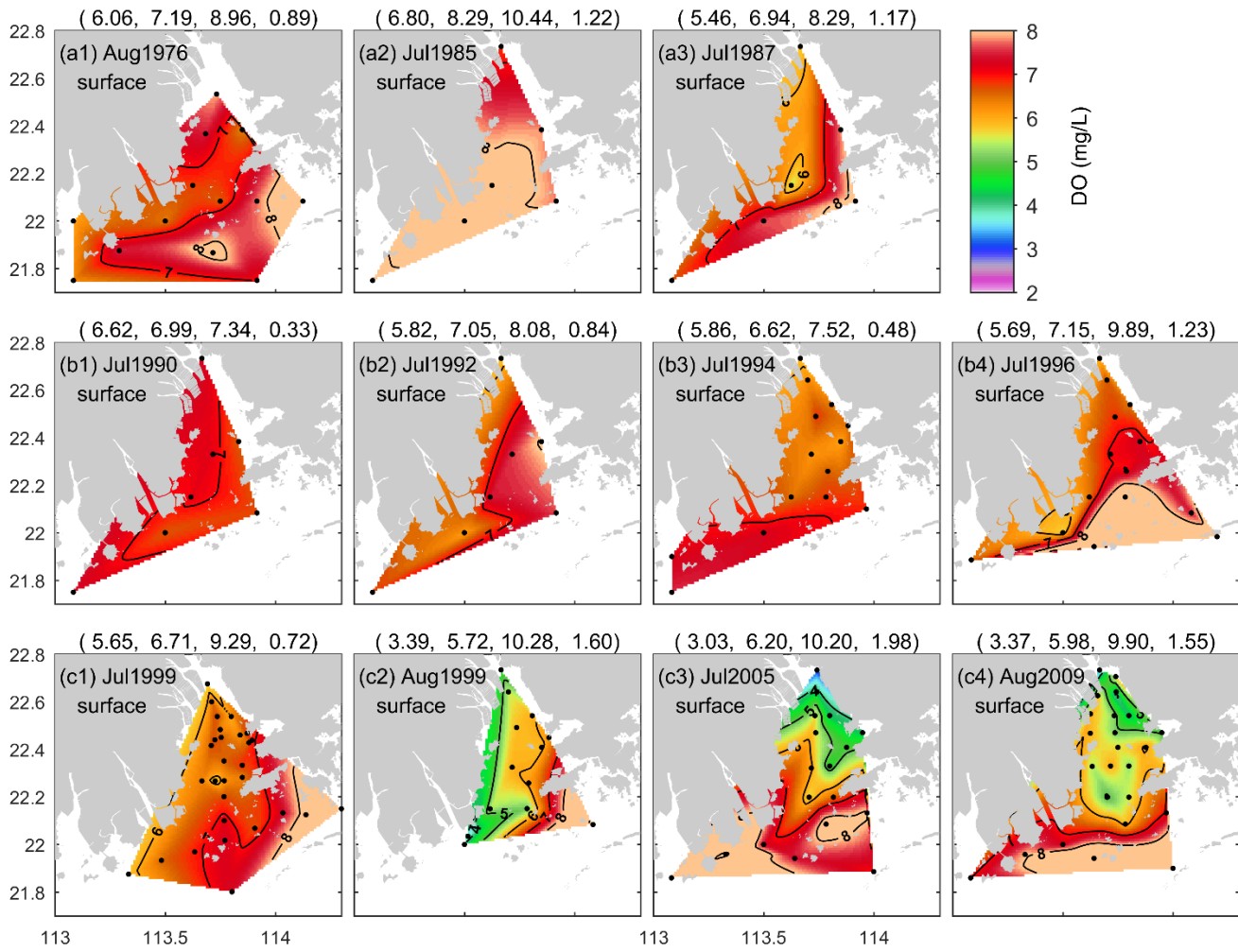


**Figure 5**. DO distributions at the surface of the PRE in summer.

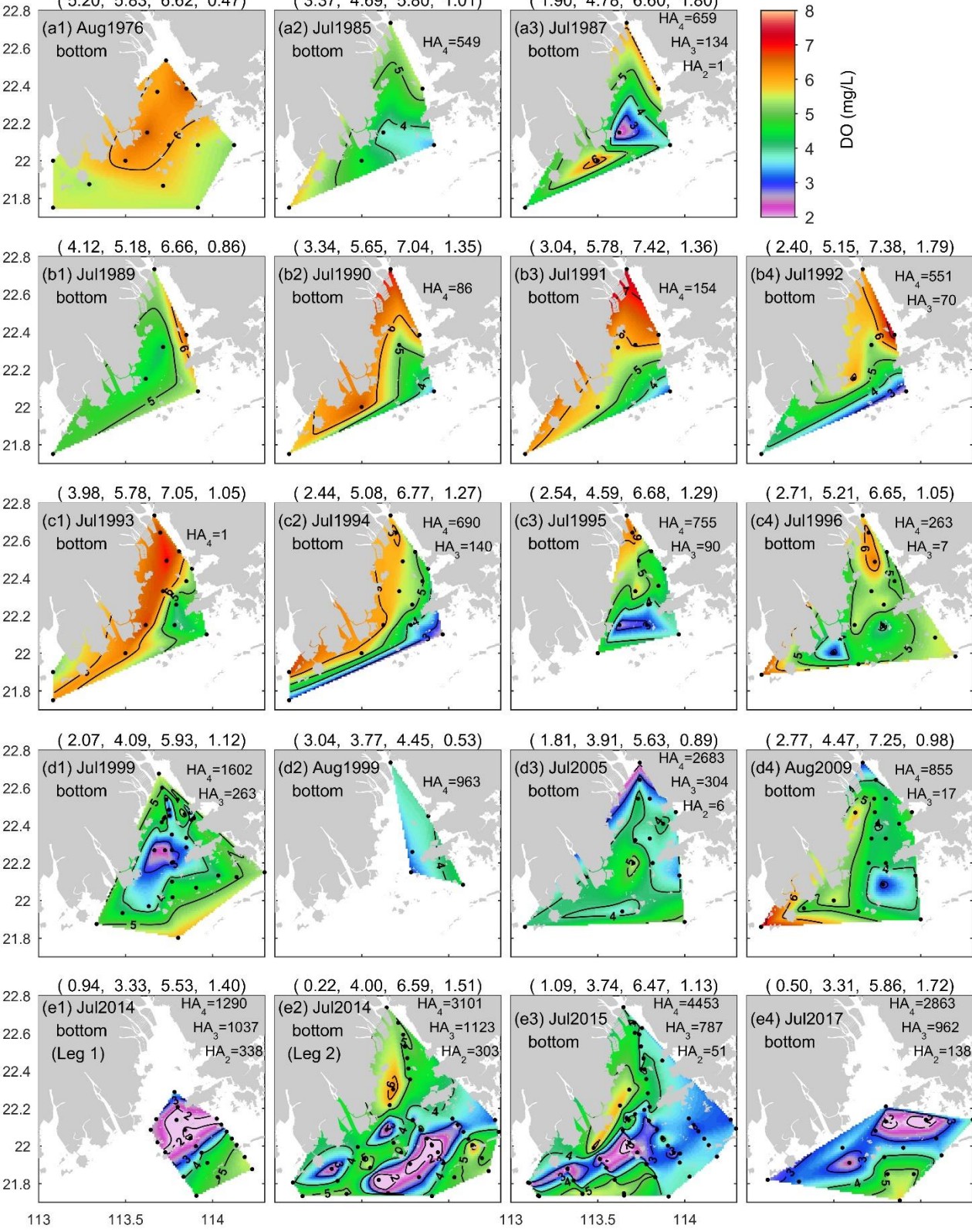

**Figure 6**. DO distributions at the bottom of the PRE for the summer months of 1976-2017. Due to the influence of Typhoon Rammasun, the cruise in July 2014 was divided into two legs, including (e1) Leg 1 in 13-16 July and (e2) Leg 2 in 19-27 July (Su et al., 2017). Note that $HA_4$, $HA_3$, and $HA_2$ represent the observation-based estimates of the areal extents ($km^2$) with DO $<$ 4, 3, and 2 mg/L, respectively.

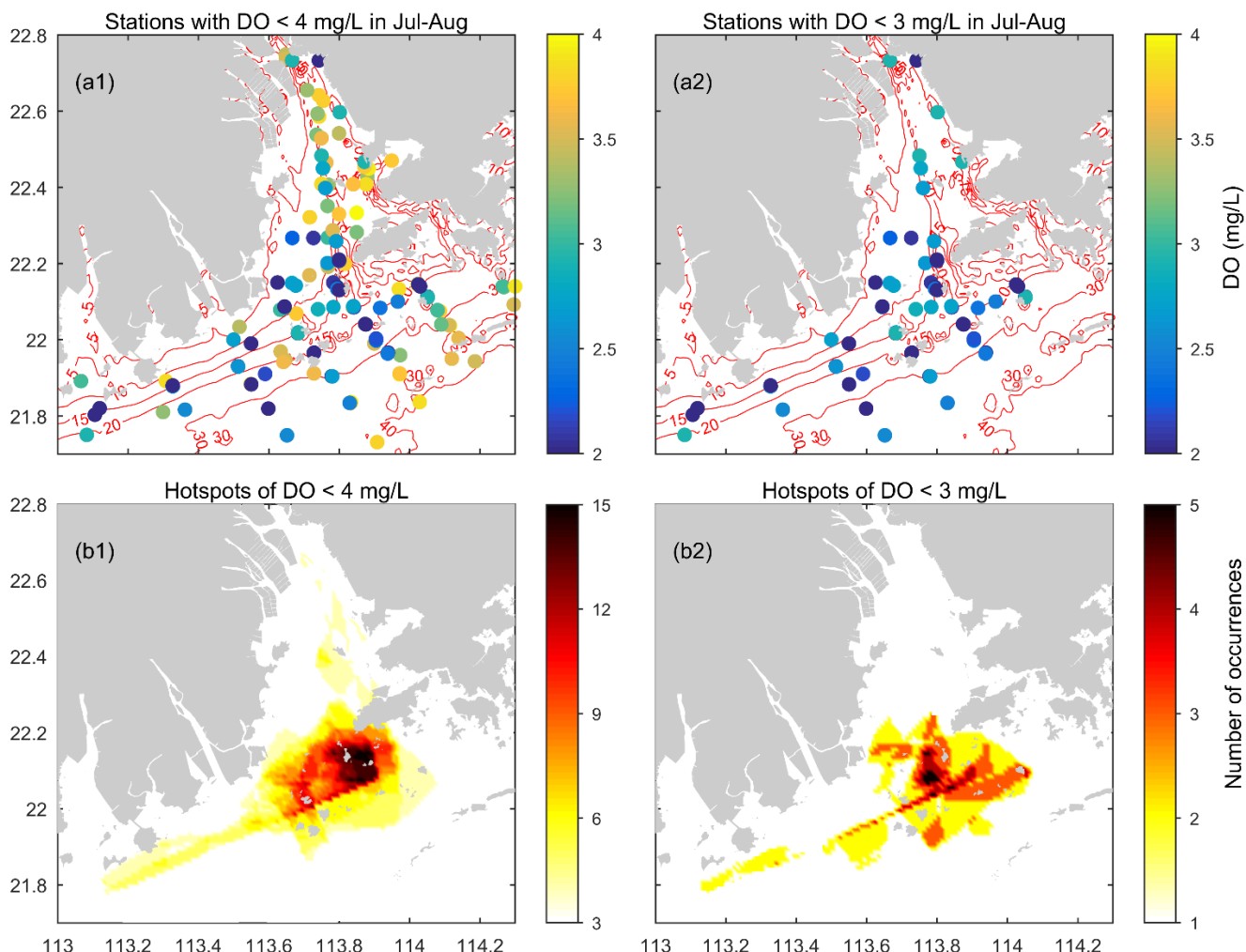

**Figure 7.** (a) The sites where low-oxygen conditions have been observed in the summer of the PRE and the corresponding lowest DO values measured (based on the data compiled from 1976 to 2017). (b) Maps showing the incidence of low-oxygen conditions at the bottom of the PRE in summer. Note that the darker color delineates higher occurrence of low-oxygen events.

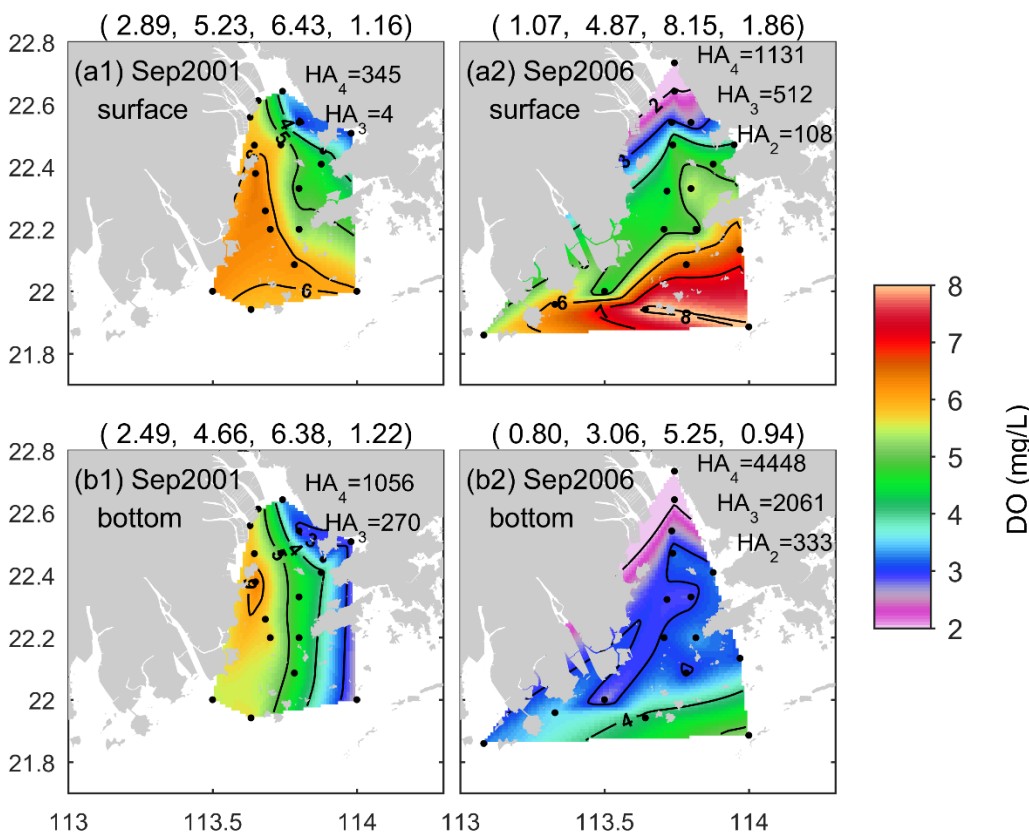

**Figure 8.** DO distributions in the (a) surface and (b) bottom waters of the PRE for the early autumn (September) of 2001 (left panels) and 2006 (right panels).


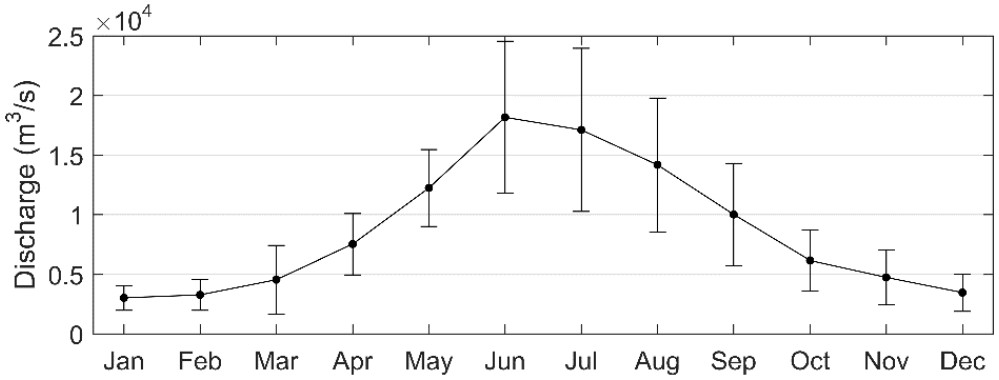

**Figure 9**. Monthly means and standard deviations of the Pearl River discharges calculated over 1979-2015.

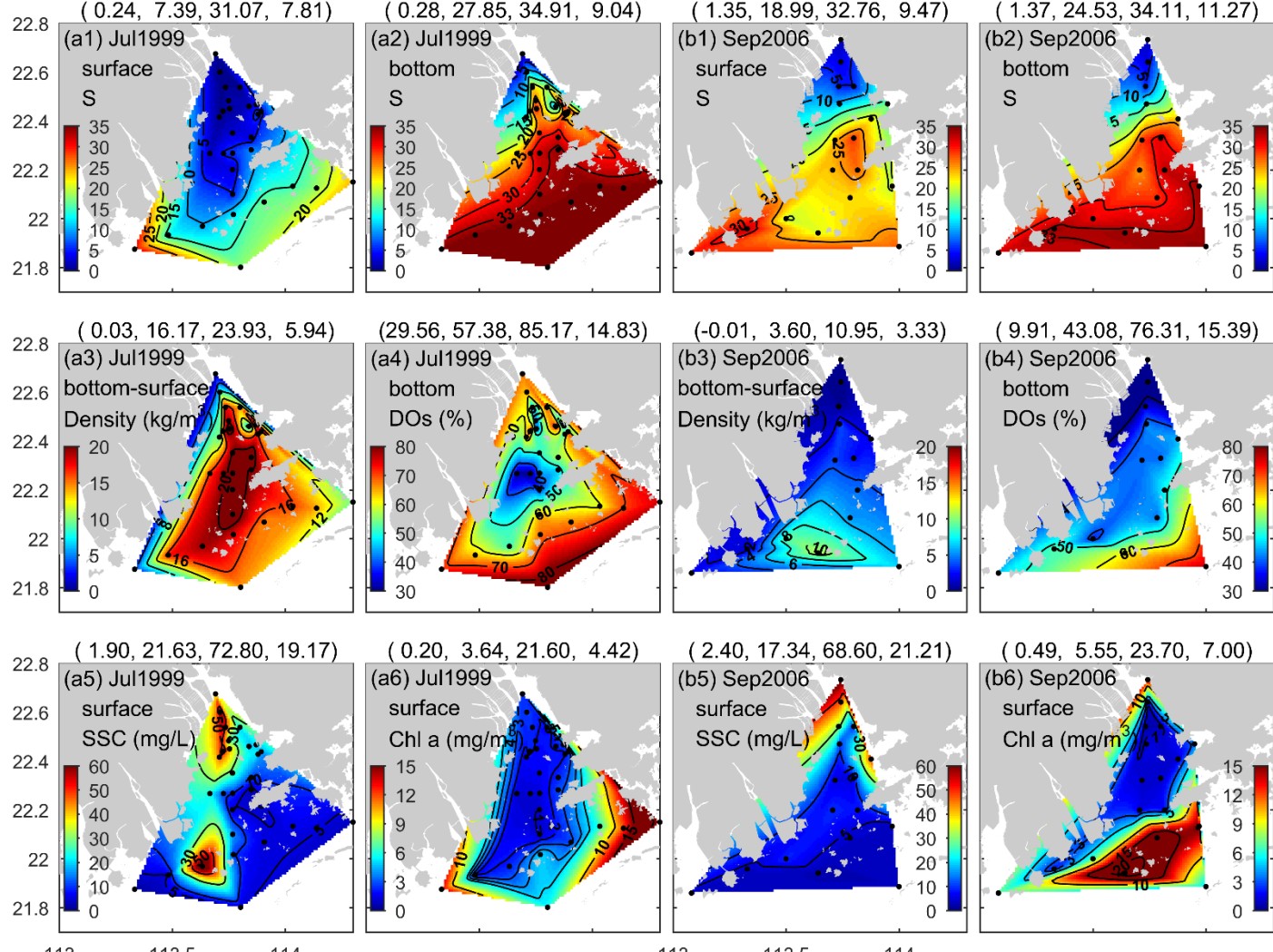

**Figure 10**. Comparison of water quality metrics observed in the PRE between (a) July 1999 (summer) and (b) September 2006 (early autumn): surface and bottom salinity (top panels); vertical density difference and bottom DO saturation ($DO_s$) (middle panels); surface suspended sediment concentrations (SSC) and chlorophyll (chl) *a* contents (bottom panels). Note that the numbers in brackets in the titles are the minimum, mean, maximum, and standard deviation values of the associated water quality metrics in sequence.

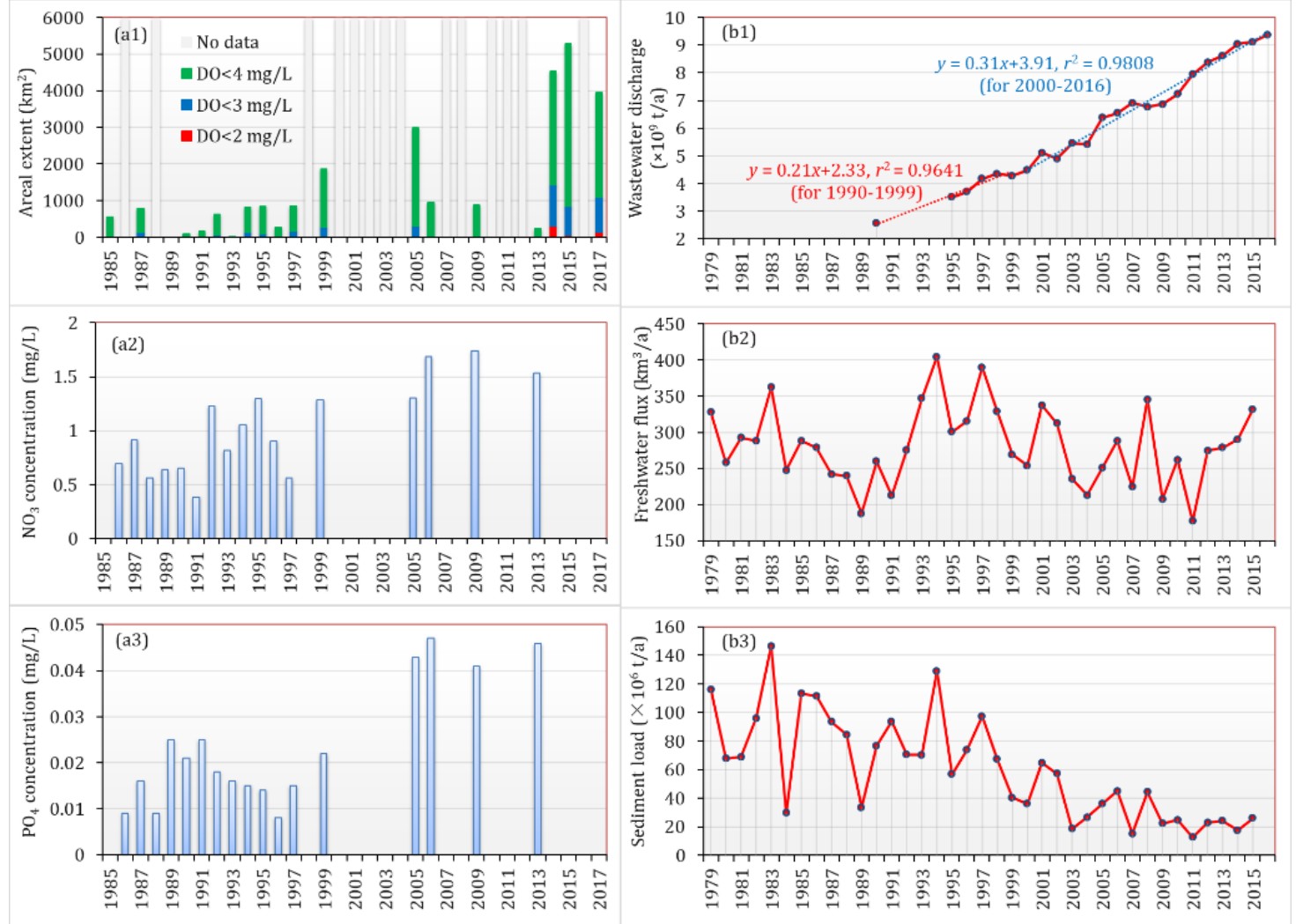

**Figure 11**. (a1) The estimated area extents of low-oxygen conditions in the bottom waters of the PRE (note that these results were estimated based on the data available so far and may have underestimated the actual ones in certain periods due to lack of sufficient observations) and (a2) the $NO_3$ and (a3) $PO_4$ concentrations near the eastern four outlets in summer during 1985-2017. (b1) Annual wastewater discharge in Guangdong Province during 1990-2016. The data before 1998 were taken from X. Li et al. (2020), and the remaining data were obtained from the Environmental Statistics Bulletin published by the Department of Ecology and Environmental of Guangdong Province (http://gdee.gd.gov.cn/tjxx3187/index.html). (b2) Freshwater discharge and (b3) sediment load of the Pearl River from 1979 to 2015, adopted from Wu et al. (2020).