# Peer review of "Long-term spatiotemporal variations and expansion of low-oxygen conditions in the Pearl River estuary: A study synthesizing observations during 1976-2017"

_Biogeosciences, 2020_

## Referee Comment (RC1)

Hu and his/her colleagues use 1976-2017 pearl river estuary monitoring data to present a analysis of PRE oxygen depletion history assessment work. The strength of this work, in my eye, includes two points. One is the 42 years of historical data itself, and the other is fig.7b, namely the statistics of hypoxic sites. The weakness of current work is also very apparent. There is a lack of deep exploration for the mechanism of the occurrence of the hypoxia, or oxygen depletion. Also, I think authors can go even further in the ststistics job. To improve the current work, I have the following comments and suggestions.

**Main:**
what is the scientific question of this work?

Why use a very old equation (Hyer et al 1971) to calculate DOsat? (line 120). Why not try the newer one? See Garcia and Gordon 1992. (Garcia and Gordon, 1992)

Authors use several independent field investigation results., but without indicating the data quality control result. Can these data be directly compared? What is the offset between variouis data set? How the water sample was collected on board? In lab what is the DO measuring method and corresponding quality control?

Why the surface water can be low in DO? Sometimes surface water can be hypoxic (line 280-285). Why? There is some work talking about this feature (surface water hypoxia) and authors should cite. Search for work by MH Dai, and /or WD Zhai.

Go deeper in statistics: authors should compare pearl river discharge history, pearl river basin GDP or fertilizer ultilization history, or any other available anthropogenic data, with their oxygen data for the period 1976-2017. In some place in the main text authors mentioned a few about this (e.g., line 355), but that's not quantitative, instead very shallow. Try compare oxygen and anthropogenic activities by numbers, and present readers by scatter plots. That would be more strong, straightfoward, and perswasive. I see authors already show some historical data (fig. 11) along. That's good. I encourage authors further incoporate oxygen data into the same plot and seek for some pattern or relation. In the main text, authors repeatedly mentioned about some threshold time point (sediment load 1999 at line 378; nutrients 2000 at line 363 et al), so I suggest try compare the oxygen data and corresponding historical data if possible.

Instead of showing grid contour maps, it is clearer by using scatter plot to show readers how the oxygen minimum value, as well as hypoxic (or low oxygen) area and oxygen depletion amount, varied from year to year.

The autumn oxygen depletion event is highlighted by the authors. But how that came into being? I am curious that authors mentioned a 'distinct mechanisms' for this autumn oxygen depletion. So, what is the mechanism? In the main text it explained as 'intricate coupling of

physical and biogeochemical processes', that does not quench readers' thirst. Oxygen depletion occurs under stratification and organic matter decay. What do authors mean by saying 'intricate coupling of physical and biogeochemical progress'? Some noval mechanisms identified in this autumn event, rather than stratification and organic matter decay? I would like to know that.

Authors suggest the peal river decreased its sediment load in recent years. And a decrease in riverine sediment load result in better light condition in the PRE, so better phytoplankton grwoth and hence worsen bottom hypoxia. While the logic sounds good, authors are suggested to do some quantitative explorations. For example, I see that authors have the suspended sediment concentration (ssc) data for dataset 1 (table 1). So what is the contour distribution patter of ssc in PRE? Maybe it matches well with the surface DO or bottom DO? What is the scatter plot if plot ssc with oxygen data?

**Minor:**
Hard to follow: line 27-29. What is the meaning of this sentense saying low oxygen area? If only read abstract, readers have no idea what is low oxygen. 3mg/L? 2mg/L? 4mg/L? and what is 'distinct mechanisms'?. If readers only read abstract, it is confusing.
Again, 'low oxygen' is mentioned in the main text (introduction line 86-87, without definition. It is really confusing to say low oxygen in a hypoxia work. Withou definiation, water beneath pycnocline can always be called low oxygen, as it is not saturated. If this is the case, then the 'low oxygen' means nothing serious in the PRE, since it is very common in many places worldwide. I see in line 99-100 there seems a low oxygen definition, it comes a little late than expected. Also I have no idea if this definiation at line 99 can be applied to term 'low oxygen' prior to this line.

Changjiang diluted water, instead of Yangtze river diluted water, is recommended. Line 63.

I am not a physical oceanographer, but for salinity, usualy no unit is needed. See line 121, line 128 and et al.

Section 3.1: What is the depth of 'bottom'? what is the sampling stratigy for the cruises during 1976-2017? How the water was collected, via what sampler? How the DO was measured?

Line 240 temperature also enhance stratification.

Fig2-8,   fig10. Too hard to read. Too small fonts.
Fig7a is confusing. It is surface or bottom? What is the meaning of color bar? Why the dots color conflicts with color bar?

Data availability: it is better to upload authors data into a public data storage, instead of share upon request. But that also depends on the data policy of the local government I guess.

Garcia, H.E., Gordon, L.I., 1992. Oxygen solubility in seawater: Better fitting equations. Limnology

and Oceanography 37, 1307-1312.

---

## Referee Comment (RC2)

**Overall quality of the preprint**

The low oxygen condition in the Pearl River Estuary has been frequently happened due to large inputs of freshwater, nutrients, and diverse contaminants from the Pearl River in recent years. With the rapidly growing population and socio-economic development at the Guangzhou, Shenzhen, and Hongkong Great Bay Area, the problem aroused many scientific community and government attentions. There has been a lot of studies on the low oxygen zone using observational data and a variety model. However, most of them were focused on short-time scale events and the associated controlling mechanisms. As far as I know, the only long-term trend study was Qian et al. (2018), but the discussion was only limited to one monitoring station south of Hong Kong rather than the entire Bay. The paper collected over four decade of cruise observations to investigate spatiotemporal variability of low oxygen condition in PRE to investigate the long-term low oxygen condition variability. It also reported that an early Autumn hypoxic event in the year 2006 and revealed the controlling mechanisms. The work is noval and the story is interesting. The manuscript is well written, flows well from topic to topic, is clear and understandable. It also structured well and the figures presented can back up the conclusion reached. I suggest acceptance after a moderate revision after considering the following points.

My major concern of the work is the inconsistency in data sampling for the long-term hypoxic area variability reported. The multi-year cruise data were not at closer stations like Gulf of Mexico or Chesapeake Bay. For example, Aug 1999 (Figure 6d2) had only five data in the Lingding Bay. All data in July 2017 are outside the Bay (Figure 6e4). This bring a problem that the area number (HA2, HA3, HA4) are lack of consistency between years. One suggestion here is putting all stations together, and finding ways to derive an oxygen number for no observation stations, and then do the calculation again. There are many of research papers for interpolation method to generate hypoxia area/volume in the Gulf of Mexico and Chesapeake Bay. The authors can introduce one of them to remedy the data inconsistency issue in the research.

Another concern of me is the early autumn low oxygen condition. To me, it seems only exist in September 2006, not other years. It should be careful for the conclusion that hypoxia undergoing a transition from episodic to seasonal regarding the time scale.

Lastly, I would expect to see a discussion about comparing long-term variability hypoxia study with other systems, like Chesapeake Bay and Gulf of Mexico.

**Specific comments**

**Line 98-Line 101:** the measure of low oxygen condition (< 2 mg/L, 3 mg/L and 4 mg/L) should be placed in the material and method section. The potential ecological consequence should also be mentioned.

**Line 116-Line 120:** Using DO saturation state as one of the low oxygen condition measure. The meaning of the new metrics should be better stated. It will be better to state how the PRE hypoxia is different from the Chesapeake Bay and Gulf of Mexico system; therefore, different measure was taken in the research

**Section 3.1 and Figure 2:** Why not think about show AOU in the analysis?

**Line 148:** "The existence of hypoxic events in periods other than summer". The statement was kind of misleading. It seems it only happened in September 2006, not something unified exist. Please emphasize and rewrite.

**Line 165:** "the observed areas" and the following area number reported. The software used for the plots, and interpolation method to generate the low oxygen area should be well reported in the method section

**Line 175:** I am confused about the statement "of which 1997, 2006 and 2013 have been shown earlier and will not be repeated here" please rewrite and clarify

**Line 180:** This is a very interesting phenomenon reported. Figure 11a should be cited here also.

**Line 266-269**: The explanations of Figure 7b1 and 7b2. This was also because of the convergence induced by cyclonic vortices in the coastal transition zone (CTZ). Please add some discussions.

**Section 4. Discussion.** I would expect to see a discussion on comparing long-term trend hypoxia variability with other systems, including both Chesapeake Bay and Gulf of Mexico. Please add section in this part.

**Table 2:** The definition of Pearson correlation coefficient should be explained in the method section. The correlation with NH4, NO3, PO4, is it with the nutrient concentration or with the loading? The details like this should be provided.

**Figure 10:** why the comparison was done between July 1999 and Sep 2006 in this figure? different year and different season. The pure bottom dissolved oxygen concentration should also be placed along with other variables

**Figure 11:** Please provide a nutrient loading figure along with other variables.

---

## Author Comment (AC1)

We wish to thank the referees for the constructive comments and suggestions which are helpful to the revision of our manuscript. Detailed response to all comments are given below (responses are shown in blue)

**General Comments**

Hu and his/her colleagues use 1976-2017 pearl river estuary monitoring data to present a analysis of PRE oxygen depletion history assessment work. The strength of this work, in my eye, includes two points. One is the 42 years of historical data itself, and the other is fig.7b, namely the statistics of hypoxic sites. The weakness of current work is also very apparent. There is a lack of deep exploration for the mechanism of the occurrence of the hypoxia, or oxygen depletion. Also, I think authors can go even further in the ststistics job. To improve the current work, I have the following comments and suggestions.

Response: Thank you for providing these comments. First of all, we would like to emphasize the significance of our study and its implication. For a long period of time, the problem of low oxygen and hypoxia in the Pearl River estuary (PRE) has attracted great attention. There have been a large number of observational and modeling studies on the low-oxygen conditions in the region, but most of them focused on short-term hypoxic events with limited data span, and there is still a lack of understanding of the long-term temporal and spatial variability of low-oxygen conditions in the PRE. Therefore, the contribution of this work is not merely on collecting the historical oxygen data itself, but more importantly, is on synthesizing these field observations during 1976-2017, for the first time (to the best of our knowledge) to attempt to elucidate the long-term evolution of low-oxygen conditions in terms of areal extents for the PRE. Specifically, our study explored the seasonal and interannual variations of oxygen status and their changes over the past 4 decades, and have revealed several important aspects on the low oxygen and hypoxia, such as the hotspot area prone to subsurface low-oxygen events, the exacerbation of summertime low-oxygen conditions, the potential transition of the PRE from a system characterized by episodic, small-scale hypoxic events to a system with seasonal, estuary-wide hypoxic conditions, etc. We believe that this work is an important supplement to the understanding of decadal changes in low-oxygen conditions in river-dominated coastal systems (like the PRE) in the context of global oxygen declines. Furthermore, this study also reported prominent hypoxic events in the early autumn of the PRE and would serve as a critical reminder for the community to realize the importance and severity of the low-oxygen problem in this period, which has long been ignored.

Secondly, it is also one of our main objectives to clarify the mechanisms and key factors controlling the occurrence of low-oxygen conditions and their expansions over recent years. We have provided some proper discussions on this by utilizing the data available to us so far and incorporating relevant findings from previous studies as well. However, to fully utilize the sparse observations, it is inevitable to use data collected from independent field surveys conducted by

different institutions with different research purpose. Some problems inherent in the observational data used limit us to make more direct comparisons and quantitative analysis. For example, as we mentioned in section 4.4 of our manuscript, there existed data gaps in certain years and lack of conformity in observational coverage, and the observations were under sampled in some years, especially before the 2010s. Besides, the available amounts of different data types are also different; for instance, the historical data on dissolved oxygen (DO), nutrient concentrations, temperature, and salinity are relatively abundant, while the long-term data on chlorophyll and nutrient loadings are lacking for us. Currently, we only have chlorophyll data in July 1999 and September 2006 on hand (as listed in Table 1 of our manuscript). Therefore, while maximizing the use of available data for analysis, we are also very cautious about its results and try to avoid the over-interpretation of these results, including the quantification of the estimated low-oxygen areas and its long-term trend. Despite the data limitations, the long-term observations show that the DO content in the PRE had significant temporal variability and spatial heterogeneity. A distinct exacerbation of low-oxygen conditions in summer could be evidenced by the increased susceptibility to large-scale low-oxygen events, be coincident with the major environment changes, and be supported by previous similar findings. These results emphasize the importance of conducting estuary-wide surveys to collect extensive data on DO and its related factors in a consistent way. In addition, this work will initiate our further studies to quantify the long-term oxygen changes and the associated mechanisms by collecting more observations to fill the data gaps as well as combining them with numerical models and/or machine learning techniques in the future.

Lastly, we agree with the reviewer's comment on providing additional statistical analysis that could be useful for linking the long-term expansion of low-oxygen conditions with the environment changes in the Pearl River region. Based on the reviewer's suggestions, we will incorporate the estimated areas of low-oxygen conditions in the PRE and the changes in nutrient concentrations along with anthropogenic activities, river discharge, and sediment load into the same figure (i.e. Figure 11 in our manuscript), and also add new scatter plots of the oxygen data versus suspended sediment concentrations (SSC). Please see our responses below for details.

**Major comments:**

1. what is the scientific question of this work?

Response: It is clearly denoted by the title of our manuscript as well as in the abstract and introduction sections that the main scientific question for our study is on the long-term spatiotemporal variations and expansion of low-oxygen conditions in the PRE and the associated key factors. Specifically, this work highlights (1) an apparent expansion of the areas affected by low oxygen in the bottom waters of the PRE during summer, which is primarily attributed to the exacerbated eutrophication associated with anthropogenic nutrient inputs and sharp decline in sediment load, (2) prominent low-oxygen events in the early autumn of the PRE, which were comparable to the most serve ones observed in summer and formed by unique mechanisms from the summer (please see our response to the Minor Comment 1 for details of the mechanisms) , and

(3) the potential transition of the PRE from a system characterized by episodic, small-scale hypoxic events to a system with seasonal, estuary-wide hypoxic conditions in summer.

As we mentioned in our response to the General Comments, we are fully aware of the limitations of the observational data in use, which largely limits our ability to quantify the long-term oxygen changes. Nevertheless, inferring from the available data, our findings on the declining trend of bottom-water DO and spatial expansion of low-oxygen conditions in the PRE are reliable from a qualitative point of view and have also been supported by previous studies (Ye et al., 2012; Qian et al., 2018). Please see lines 347-355 of our manuscript for further details.

2. Why use a very old equation (Hyer et al 1971) to calculate DOsat? (line 120). Why not try the newer one? See Garcia and Gordon 1992.(Garcia and Gordon, 1992)

Response: Thank you for the comment. As suggested, we used the newer equation proposed by Garcia and Gordon (1992) to re-calculate the oxygen saturation concentrations (DOsat). The new results (shown in Figure r1 below) are close to the original ones (their relative differences are mostly within 2%). The main findings remain solid. We will revise the equation, the figure and related numbers in our manuscript accordingly.

[Figure]

Figure r1. Spatial means and standard deviations of DO concentrations, DO saturation (DOs), apparent oxygen utilization (AOU), salinity (S), and temperature (T) in the surface and bottom

waters of the PRE in (a) spring (March-May) and winter (December-February), (b) summer (June-August), and (c) autumn (September-November) during 1976-2014. Note that the red dots in the first row of the figure represent the lowest DO values measured in each time period.

3.  Authors use several independent field investigation results., but without indicating the data quality control result. Can these data be directly compared? What is the offset between variouis data set? How the water sample was collected on board? In lab what is the DO measuring method and corresponding quality control?

Response: As we mentioned in our response to the General Comments, the spatiotemporal variations and long-term evolution of low-oxygen conditions in the PRE are poorly understood at the current stage. One major reason is the lack of accessible continuous observations for oxygen and a synthesis of relevant historical data (note that the previous studies on low oxygen and hypoxia in the PRE mostly focused on short-term events with limited data span). For us, with the aim to advance the research progress on the long-term oxygen changes in the PRE, our strategy is to make full use of a variety of data sources to integrate all available observations as far as possible. Thus, it is inevitable to use data collected from independent field surveys conducted by different institutions. We totally understand the reviewer's concern about the data quality control and their comparability. In fact, in order to minimize the uncertainties of the data, we selected data only from reliable sources with formal publication and usage records, which would ensure the reliability and quality control of the data.

The observational data we collected involve five datasets compiled from different sources. Specifically, as denoted in Table 1 of our manuscript, the first dataset (Dataset 1) includes water quality observations from 42 cruises during 1976-2006 conducted by the South China Sea Environmental Monitoring Center. Part of the data were also used for analysis by Li et al (2020), in which the methods of sampling and chemical analysis were described in their section 2.1. The sample collection, storage and transportation, seawater analysis, and data processing and quality control were strictly operated in accordance with the specifications of oceanographic survey (e.g., GB/T 12763-1991 and GB/T 12763-2007) and the specifications for marine monitoring (e.g., GB 17378-1998 and GB 17378-2007) issued by the National Standard of P.R. China. By following these specifications, three-point samples were collected from the surface (0.5 m below the sea surface), half depth, and bottom (0.5-2 m above the sea bed) when the water depth was > 10 m; two-point samples were collected from the surface and bottom when the depth was between 5 and 10 m; and only surface sample was collected when the depth was < 5 m. Temperature was measured on board using a thermometer, and salinity was measured with an induction salinometer in the laboratory. Ammonia ($NH_4$), nitrate ($NO_3$), and phosphate ($PO_4$) were analyzed using the indophenols blue spectrophotometric, Cd reduction, and phosphorus molybdenum blue spectrophotometric methods, respectively. Suspended sediment concentrations (SSC) were measured by the gravimetric method, and chlorophyll $a$ was measured using a spectrophotometer after the acetone extraction. As for DO, water samples were collected in brown frost-mouth bottles,

immediately fixed with solutions of $MnCl_2$ and KI on board, and were analyzed using the Winkler titration method (Parson et al., 1984). According to the requirements of data quality control, double-parallel samples were obtained to ensure the accuracy and comparability of the sample measurements.

The third dataset (Dataset 3) with observations for 4 seasonal cruises during 2006-2007 and the fourth dataset (Dataset 4) with observations for 4 seasonal cruises during 2013-2014 both followed the same specifications as for Dataset 1 in terms of sampling procedures and chemical analysis. It is important to note that although the specifications issued by the National Standard of China have changed over time, the methodology and fundamental principles for analyzing salinity, DO, nutrients, and chlorophyll involved in this work have not changed, ensuring the accuracy and comparability of the data.

With respect to Dataset 2, the observations were collected from a summer cruise conducted by the Pearl River Estuary Pollution Project in July 1999 (Chen et al., 2004). The vertical profiles for temperature, salinity, turbidity, DO, and chlorophyll *a* were measured using a YSI-6600 multi-parameter automatic water quality sensor. The instrument was calibrated twice with standard samples. The chlorophyll *a* data obtained were compared with those obtained from 169 water samples measured by Turner Designs 10-005R fluorescence method, and the DO content was calibrated against the saturation level prior to each profile measurement (Yin et al., 2004). As for nutrients, samples were collected by Go-flo water samplers from the surface, middle, and bottom, and were measured on board with traditional standard methods following the same specification as for the datasets mentioned above. The physical and biochemical parameters of Dataset 2 have been used in multiple observational studies (e.g., Yin et al., 2004; Yang et al., 2011) and modelling studies (e.g., Hu et al.,2009; Luo et al., 2009).

Regarding Dataset 5, it was comprised of recent observations on bottom-water DO data collected in July of 2014, 2015, and 2017 reported by Su et al. (2017), Lu et al. (2018), and Shi et al. (2019), respectively. All these DO data were measured on board using the classic Winkler titration method (Parson et al., 1984). Please see the Materials and methods sections in the corresponding literatures for more details.

Based on the reviewer's comments, we will provide supplementary details of the corresponding sampling procedures and chemical analysis involved in the five datasets. Also, we will provide further explanations on the quality control of the data in use and their comparability. It should be mentioned that although we cannot fully eliminate the potential data inconsistences, which is inevitable, this work has a significant contribution to advancing our understanding on the long-term variability and expansion of low-oxygen conditions in the PRE, and it also serves as an important reminder for the community to conduct estuary-wide field investigations in a consistent way.

4. Why the surface water can be low in DO? Sometimes surface water can be hypoxic (line 280-285). Why? There is some work talking about this feature (surface water hypoxia) and authors

should cite. Search for work by MH Dai, and /or WD Zhai.

Response: Based on the long-term observations, we found that the low-oxygen water frequently appeared in the surface waters of the inner Lingdingyang Bay in recent years, as also reported by previous studies (Zhai et al. 2005; He et al., 2014; Li et a., 2020). This phenomenon was primarily attributed to the influence of low-oxygen inflows from the upstream reaches as a result of intense nitrification and aerobic respiration of organic matter from direct anthropogenic inputs (He et al., 2014 - a work conducted by Dai's lab). Please see detailed discussions in our manuscript (lines 178-183 and lines 367-370). We will add the citations of Zhai et al. (2005) and He et al. (2014) as suggested.

5. Go deeper in statistics: authors should compare pearl river discharge history, pearl river basin GDP or fertilizer ultilization history, or any other available anthropogenic data, with their oxygen data for the period 1976-2017.In someplace in the main text authors mentioned a few about this (e.g., line 355), but that's not quantitative, instead very shallow. Try compare oxygen and anthropogenic activities by numbers, and present readers by scatter plots. That would be more strong, straightfoward, and perswasive. I see authors already show some historical data (fig. 11)along. That's good. I encourage authors further incorporate oxygen data into the same plot and seek for some pattern or relation. In the main text, authors repeatedly mentioned about some threshold time point (sediment load 1999 at line 378; nutrients 2000 at line 363 et al), so I suggest try compare the oxygen data and corresponding historical data if possible.

Response: We agree with the reviewer that it will help us to further explore the link between the long-term expansion of low-oxygen conditions and the environment changes in the Pearl River region by incorporating the oxygen data, anthropogenic activities, river discharge, and sediment load into the same figure. Accordingly, we added the estimated areas of low-oxygen conditions in the bottom waters of the PRE during 1985-2017 and the nutrient concentrations near the eastern four river outlets along with the wastewater discharge to reflect the pressure of anthropogenic pollutant inputs (please note that the long-term nutrient loadings are not available) into Figure 11 of our manuscript. Please see the revised figure (Figure r2) below.

[Figure]

Figure r2. (a1) The estimated area extents of low-oxygen conditions in the bottom waters of the PRE and the (a2) $NO_3$ and (a3) $PO_4$ concentrations near the eastern four outlets in summer during 1985-2017. (b1) Annual wastewater discharge in Guangdong Province during 1990-2016. The data before 1998 were taken from Li et al. (2020), and the remaining data were obtained from the Environmental Statistics Bulletin published by the Department of Ecology and Environmental of Guangdong Province (http://gdee.gd.gov.cn/tjxx3187/index.html). (b2) Freshwater discharge and (b3) sediment load of the Pearl River from 1979 to 2015, adopted from Wu et al. (2020).

The above figure clearly shows a distinct exacerbation of summertime low-oxygen conditions as the increased frequencies in extremes, and an increasing trend in the nutrient concentrations along with the wastewater discharge. Although there existed data gaps in certain years, it is still clear that the nutrient concentrations after 2000 are higher than those before. This finding is also supported by Li et al. (2020), which found that the nutrient concentrations in the upstream reaches mostly exceeded 50 μg/L for $NH_4$, 1000 μg/L for $NO_3$, and 30 μg/L for $PO_4$ since 2000 by analyzing the 24-year time series data obtained during 1988-2011.We have cited their findings regarding the changes in nutrients in our manuscript (lines 361-364). In addition to the rise in nutrients, the sediment load of the Pearl River (data adopted from Wu et a. (2020)) experienced a significant decline from 1979 to 2015, while the freshwater discharge only showed a slight

declining trend. This is consistent with the findings by Wu et al. (2020); they investigated the sediment load of the nine major rivers in China (including the Yangtze, Pearl, and Yellow rivers) and found that the sediment load has dramatically dropped by 85% over the past 6 decades, and they also found from the statistical analysis that the year 1999 was one of the important time nodes for the sediment decline. We have also cited their findings regarding the changes in the sediment load in our manuscript (lines 376-378). Based on our calculation, the sediment load of the Pearl River was approximately dropped by 63% between 1979-1998 and 1999-2015. Such an abrupt change, superimposed with the changes in nutrients, would act on the expansion of low-oxygen conditions in the PRE. Nevertheless, further studies are needed to clarify the role and relative contributions of these changes in the long-term trend of low-oxygen conditions in the PRE by combining the observations with numerical models.

Finally, we would like to emphasize that cautions should be kept when interpreting the changes of the low-oxygen areas as shown in Figure r2. These results, estimated from the available data so far, should not be directly used to quantify the long-term deoxygenation trend due to the data limitations as we pointed out in our response to the General Comments. In order to avoid the potential misleading or over-interpretation from the figure, we will provide necessary explanations on this issue in the figure caption and the text of our manuscript.

6. Instead of showing grid contour maps, it is clearer by using scatter plot to show readers how the oxygen minimum value, as well as hypoxic (or low oxygen) area and oxygen depletion amount, varied from year to year.

Response: Compared with the northern Gulf of Mexico and the Yangtze River estuary, the low-oxygen zone in the PRE shows relatively significant temporal and spatial variability, with locations and severity varying greatly from year to year (Figures 3-6 in our manuscript). Therefore, in order to comprehensively present the occurrence and spatial patterns of the low-oxygen conditions in different years, we chose to use the contour maps. In addition, we did use scatter plots to show the oxygen minimum values (please see the red dots in the first row of Figure 2 of our manuscript) and the oxygen saturation state (DOs). Please note that we have added new subplots of apparent oxygen utilization (AOU, indicating the oxygen depletion amount) into the same figure (please see Figure r1 above) as suggested by the reviewer #2.

7. The autumn oxygen depletion event is highlighted by the authors. But how that came into being? I am curious that authors mentioned a 'distinct mechanisms' for this autumn oxygen depletion. So, what is the mechanism? In the main text it explained as 'intricate coupling of physical and biogeochemical processes', that does not quench readers' thirst. Oxygen depletion occurs under stratification and organic matter decay. What do authors mean by saying 'intricate coupling of physical and biogeochemical progress'? Some noval mechanisms identified in this autumn event, rather than stratification and organic matter decay? I would like to know that.

Response: As we discussed in our manuscript (section 4.2, lines 276-325), we speculate that the hypoxic and low-oxygen events in early autumn were caused by (1) the inflows of low-oxygen waters from the upstream reaches and (2) enhanced oxygen depletion driven by an intricate coupling of physical and biogeochemical processes. Firstly, in early autumn the freshwater discharge has decreased to about 60% of the summertime discharge (10,000 m³/s). This would reduce the intensity of two-layers gravitational circulations in the Lingdingyang Bay, i.e. the weaker offshore extension of fresh water at the surface layer and milder onshore intrusion of saline water at the bottom layer (Figure 10 in our manuscript). As a result, the low-oxygen freshwater from the upstream reaches could be transported further into the Lingdingyang Bay at the bottom layer. This can be supported by the high correlation between the oxygen and salinity as shown in Table 2 of our manuscript. Secondly, the reduced freshwater discharge would also facilitate the settling down of terrestrial organic carbon within the Lingdingyang Bay. The thereafter respiration of these organic carbon could also maintain the low oxygen and even hypoxic levels within the bay. In addition, the light availability would be largely improved, which in combination with the prolonged residence time would favor the primary production and ultimately the oxygen consumptions due to locally produced organic matter.

The intricate coupling of physical and biogeochemical processes is a summarization of all the processes that we have discussed above in the same section of our manuscript (section 4.2). Specifically, they include the facilitated deposition of terrestrial organic carbon, the increased light availability, the prolonged residence time, the enhanced primary production, and etc.

In a summary, the mechanisms of early-autumn hypoxia appear to be different from that of the summer one. The inflow of low-oxygen freshwater is the first-order mechanism and the organic matter decay can maintain the hypoxia within the Lingdingyang Bay. With respect to stratification, as we mentioned in this section of the manuscript (lines 292-296), it is not as important as in the summer because there is no significant correlation between the bottom oxygen and the vertical density gradient (Table 2 in our manuscript).

8.  Authors suggest the peal river decreased its sediment load in recent years. And a decrease in riverine sediment load result in better light condition in the PRE, so better phytoplankton grwoth and hence worsen bottom hypoxia. While the logic sounds good, authors are suggested to do some quantitative explorations. For example, I see that authors have the suspended sediment concentration (ssc) data for dataset 1 (table 1). So what is the contour distribution patter of ssc in PRE? Maybe it matches well with the surface DO or bottom DO? What is the scatter plot if plot ssc with oxygen data?

Response: With respect to the impact of the sediment decline on the oxygen changes, please see our response to the Major Comment 5 for detailed discussions. In brief, the sediment load of the Pearl River experienced a significant decline from 1979 to 2015, which is consistent with the findings by Wu et al. (2020). It was approximately dropped by 63% between 1979-1998 and 1999-2015. We have shown the distributions of SSC in the PRE during July 1999 and September 2006

(Figure 10 in our manuscript), and the different patterns of SSC and its effects on the growth of phytoplankton in different seasons can be observed. For the rest of the years with available SSC data, the distributions of SSC are given in Figure r3 below. It can be seen that the SSC shows a spatial pattern of being high at the nearshore and low in the offshore waters. Besides, the averaged values of SSC in the 2000s-2010s were generally lower than those in the early 1990s, which is consistent with the declining trend of the sediment load of the Pearl River.

[Figure]

Figure r3. Spatial distributions of suspended sediment concentrations (SSC) at the surface of the PRE in summer.

As suggested by the reviewer, we also plotted the SSC with oxygen data for the surface and bottom waters (please see Figure. r4 below). There is no obvious relationship between these two variables at the bottom, but there is a negative correlation at the surface, that is, the higher SSC, the lower oxygen. On the one hand, this implies that the physical transport and dynamic processes of suspended sediments (e.g., flocculation, deposition, suspension caused by erosion at the bottom layer), in combination with the joint effects of various physical and biochemical processes on oxygen, complicates the intrinsic linkage between suspended sediments and oxygen in the bottom waters. On the other hand, the negative correlation between SSC and oxygen at the surface suggests that with the decrease in SSC, water transparency greatly improves and thus favors the growth of phytoplankton, and thereby the surface oxygen increases as a result of the oxygen release via photosynthesis.

[Figure]

Figure r4. Oxygen versus SSC at the surface and bottom of the PRE in summer

The significant decline of the sediment load of the Pearl River, superimposed with the changes in nutrients, would act on the expansion of low-oxygen conditions in the PRE. However, we realized that the data available so far are not able to clarify the role and relative contributions of these changes in the long-term trend of low-oxygen conditions in the PRE. Therefore, further studies by combining the observations with numerical models are needed to address these important questions in the future.

According to the reviewer's suggestions, we will add the above discussions regarding the effect of SSC on oxygen in the main text of our manuscript and add the distribution patterns of SSC in different years and scatter plots of oxygen versus SSC in the supplementary materials of our manuscript.

**Minor comments:**

(1) Hard to follow: line 27-29. What is the meaning of this sentense saying low oxygen area? If only read abstract, readers have no idea what is low oxygen. 3mg/L? 2mg/L? 4mg/L? and what is 'distinct mechanisms'?. If readers only read abstract, it is confusing.

Response: In this study, we refer the oxygen concentrations below 2, 3, and 4 mg/L to as hypoxia, oxygen deficiency, and low oxygen, respectively. We will make these definitions clear earlier in our revised manuscript. As for the 'distinct mechanisms', please see our response to the Major Comment 7. Here we mean that the mechanisms for the early-autumn low oxygen events are different from that in the summer. We will change this sentence in our revised manuscript into:

*A large area affected by low oxygen (DO<4mg/L) was found in September 2006, where the low-oxygen conditions were comparable to the most severe ones observed in summer. It was formed by the inflows of low-oxygen waters from the upstream reaches and enhanced oxygen depletion driven by an intricate coupling of physical and biogeochemical processes.*

(2) Again, 'low oxygen' is mentioned in the main text (introduction line 86-87, without definition. It is really confusing to say low oxygen in a hypoxia work. Withou definiation, water beneath pycnocline can always be called low oxygen, as it is not saturated. If this is the case, then the 'low oxygen' means nothing serious in the PRE, since it is very common in many places

worldwide. I see in line 99-100 there seems a low oxygen definition, it comes a little late than expected. Also I have no idea if this definiation at line 99 can be applied to term 'low oxygen' prior to this line.

Response: Thank you for the comment. The 'low oxygen' in this study was defined as the oxygen concentrations below 4 mg/L. We will move this definition earlier in our revised manuscript to make it clearer.

(3) Changjiang diluted water, instead of Yangtze river diluted water, is recommended. Line 63.
Response: We will revise it as suggested.

(4) I am not a physical oceanographer, but for salinity, usualy no unit is needed. See line 121, line 128 and et al.
Response: We will revise it as suggested.

(5) Section 3.1: What is the depth of 'bottom'? what is the sampling stratigy for the cruises during 1976-2017? How the water was collected, via what sampler? How the DO was measured?
Response: Bottom samples were collected at the waters 0.5-2 m above the sea floor. Please see our response to the Major Comment 3 for details. We will provide supplementary information on the sampling procedures and chemical analysis involved in the five datasets we used. Also, we will provide further explanations on the quality control of the data in use and their comparability.

(6) Line 240 temperature also enhance stratification.
Response: Yes, we agree. However, in many coastal hypoxic systems (e.g., the Yangtze River estuary, Chesapeake Bay, and the northern Gulf of Mexico), salinity has a much stronger effect on stratification compared to temperature (Fennel and Testa, 2019). This is also the case for the PRE, in which the stratification is mainly determined by salinity due to large freshwater discharges (Wong et al., 2003; Hu et al., 2011).

(7) Fig2-8,fig10. Too hard to read. Too small fonts.
Response: We will revise these figures to make their fonts clearer in our revised manuscript.

(8) Fig7a is confusing. It is surface or bottom? What is the meaning of color bar? Why the dots color conflicts with color bar?
Response: It is bottom. It shows the stations where the low oxygen conditions (DO < 4 mg/L) and oxygen deficiency (DO < 3 mg/L) have been observed during July and August. The dots color represents the observed minimum oxygen concentrations, where the dark blue represents the lower oxygen concentrations and the yellow represents the relatively higher oxygen concentrations. Please note that the dots color is consistent with the color bar.

(9) Data availability: it is better to upload authors data into a public data storage, instead of share upon request. But that also depends on the data policy of the local government I guess.

Response: Please see our response to the Major Comment 3 for details on the data sources of all five datasets we compiled. Among these datasets, Datasets 1 and 3 are copyrighted and not allowed to be released in their original forms according to the administrations' data policy. As for Datasets 2 and 4, there is no such restriction and thus we will share these data through public data storage. Dataset 5 was derived from the literatures (Su et al., 2017; Lu et al., 2018; Shi et al., 2019) and can be downloaded directly via the links provided in the corresponding literatures.

References

[revised manuscript text omitted]

---

## Author Comment (AC2)

We wish to thank the referees for the constructive comments and suggestions which are helpful to the revision of our manuscript. Detailed response to all comments are given below (responses are shown in blue)

**General Comments**

The low oxygen condition in the Pearl River Estuary has been frequently happened due to large inputs of freshwater, nutrients, and diverse contaminants from the Pearl River in recent years. With the rapidly growing population and socio-economic development at the Guangzhou, Shenzhen, and Hongkong Great Bay Area, the problem aroused many scientific community and government attentions. There has been a lot of studies on the low oxygen zone using observational data and a variety model. However, most of them were focused on short-time scale events and the associated controlling mechanisms. As far as I know, the only long-term trend study was Qian et al. (2018), but the discussion was only limited to one monitoring station south of Hong Kong rather than the entire Bay. The paper collected over four decade of cruise observations to investigate spatiotemporal variability of low oxygen condition in PRE to investigate the long-term low oxygen condition variability. It also reported that an early Autumn hypoxic event in the year 2006 and revealed the controlling mechanisms. The work is noval and the story is interesting. The manuscript is well written, flows well from topic to topic, is clear and understandable. It also structured well and the figures presented can back up the conclusion reached. I suggest acceptance after a moderate revision after considering the following points.

Response: We are very grateful to the reviewer for the positive comments and recognition of the novelty and significance of this work. We will revise the manuscript as suggested.

**Major comments:**

1. My major concern of the work is the inconsistency in data sampling for the long-term hypoxic area variability reported. The multi-year cruise data were not at closer stations like Gulf of Mexico or Chesapeake Bay. For example, Aug 1999 (Figure 6d2) had only five data in the Lingding Bay. All data in July 2017 are outside the Bay (Figure 6e4). This bring a problem that the area number (HA2, HA3, HA4) are lack of consistency between years. One suggestion here is putting all stations together, and finding ways to derive an oxygen number for no observation stations, and then do the calculation again. There are many of research papers for interpolation method to generate hypoxia area/volume in the Gulf of Mexico and Chesapeake Bay. The authors can introduce one of them to remedy the data inconsistency issue in the research.

Response: We totally understand the reviewer's concern. In fact, although the summertime hypoxia in the Pearl River estuary (PRE) has been reported since the 1980s, there is still a lack of understanding of its long-term evolution. One major reason is the lack of accessible continuous

observations for oxygen and a synthesis of relevant historical data. To the best of our knowledge, this is a first study to collect the estuary-wide historical observations over 40 years and attempt to elucidate the long-term evolution of low-oxygen conditions in the PRE. Since these observations were conducted by different institutions for different scientific purposes, there is inevitable inconsistence in the sampling stations.

We would like to thank the reviewer very much for the suggestions. However, the extrapolation of oxygen data into unobserved stations will introduce large uncertainties, especially when the sampling stations are often limited and localized before the 2010s. We have fully realized the data limitations in use. As we discussed in section 4.4 of our manuscript, the data gaps in some years and the lack of conformity in observational coverage largely limit our ability to quantify the long-term changes of low-oxygen conditions in the PRE. We are also very cautious about our conclusions and try not to overinterpret them. Nevertheless, our findings on the declining trend of bottom DO and its spatial expansion in the PRE emphasize the importance of conducting estuary-wide surveys to collect extensive oxygen data in a consistent way. Based on the conclusions from our study, we would also suggest to build the estuary-wide estimates of oxygen by combining these long-term observations with the numerical models and/or machine learning systems to better quantify the long-term oxygen changes and the associated mechanisms; however, this is beyond the scope of this study.

2. Another concern of me is the early autumn low oxygen condition. To me, it seems only exist in September 2006, not other years. It should be careful for the conclusion that hypoxia undergoing a transition from episodic to seasonal regarding the time scale.

Response: It is our statement that has caused this misunderstanding. Actually, the conclusion refers to the potential transition of the summertime hypoxia in the PRE, which was deduced by the long-term observational oxygen data in summer. As pointed out by the reviewer, we also realized that this conclusion is not applicable to the hypoxic conditions in early autumn due to the data limitations in this period. Based on the reviewer's suggestion, we will revise the conclusion to make it explicitly referring to the summertime hypoxia.

3. Lastly, I would expect to see a discussion about comparing long-term variability hypoxia study with other systems, like Chesapeake Bay and Gulf of Mexico.

Response: We agree that it would be interesting to compare the long-term variabilities of hypoxia across different systems under the context of global oxygen declining. However, the main focus of this study is on the long-term variations of hypoxia in the PRE and the underlying mechanisms, and due to the data limitations that largely limit our ability to quantify the long-term oxygen changes, it is immature for us at the current stage to have an in-depth comparison between the PRE and other hypoxic systems (please note that comprehensive comparisons between other different hypoxic systems have been conducted in previous studies (e.g., Rabouille et al., 2008; Fennel and Testa, 2019)). Alternatively, we will add some moderate discussions in the first two paragraphs of

section 4.3 in our manuscript as follows:

*"Apparent long-term expansions of hypoxic conditions have been documented in several coastal systems where sustained seasonal hypoxia has been reported. For instance, the hypoxic volume in the Baltic Sea has expanded dramatically with increasing nutrient inputs from its watershed and enhanced water-column respiration resulting from warming (Fennel and Testa, 2019 and references therein). In the Chesapeake Bay, the hypoxia can be tracked back to the 1930s and has witnessed an expansion of its volume since the 1950s due to the increased nutrient loads (Hagy et al., 2004). Moreover, the hypoxia in the northern Gulf of Mexico has been documented since 1985 (Rabalais et al., 2002). However, models suggest that the occurrence of large-scale hypoxia can be as early as the 1970s (Rabalais et al., 2007). Despite large interannual variability, the hypoxic area has increased from an average of 8,300 km² in 1985-1992 to 16,000 km² in 1993-2001 (Scavia et al., 2003). To mitigate the hypoxia, nutrient reduction plans have been proposed. In addition to the nutrient loads, the long-term climate change can also exaggerate hypoxia and reduce the positive impacts from nutrient reduction. Modeling studies have suggested that the worsened physical conditions since the 1980s in the Chesapeake Bay, e.g. prolonged vertical exchange time and elevated temperature, can contribute to the increased hypoxia (Du and Shen, 2015; Du et al., 2018). A more recent study shows that the impacts from climate change and nutrient reduction cancel out and therefore the hypoxic volume in the Chesapeake Bay shows no significant long-term trends in the past three decades (Ni et al., 2020). Similar findings have been also archived in other hypoxic systems, e.g. the northern Gulf of Mexico (Kemp et al., 2009; Obenour et al., 2013). However, it has to be noticed that the susceptibility of hypoxia to increased anthropogenic activities varies across different coastal systems due to their physical and biological features.*

*It is commonly recognized that the PRE did not develop similar large-scale, persistent low-oxygen zone as in other hypoxic systems (e.g., the northern Gulf of Mexico, the Yangtze River estuary). A combination of intriguing features including shallow and turbid waters, rapid physical exchanges, and unstable vertical stratification provides good buffering capacity for the PRE to mitigate eutrophication and hypoxic conditions in summer…."*

**Specific Comments:**

(1) **Line 98-Line 101:** the measure of low oxygen condition ($< 2$ mg/L, 3 mg/L and 4 mg/L) should be placed in the material and method section. The potential ecological consequence should also be mentioned.

Response: Thank you for the comment. We will revise it as suggested. As for the potential ecological consequence, it has been mentioned in the first paragraph of the Introduction section (please see lines 41-46).

(2) **Line 116-Line 120:** Using DO saturation state as one of the low oxygen condition measure. The meaning of the new metrics should be better stated. It will be better to state how the PRE hypoxia is different from the Chesapeake Bay and Gulf of Mexico system; therefore, different

measure was taken in the research.

Response: Please note that the oxygen saturation state (DOs, %) in use, defined as the ratio of the in situ oxygen concentration to its saturation level at a known temperature and salinity, is not a new metric we propose. It has been documented in the specifications for marine monitoring (e.g., GB 17378-2007) issued by the National Standard of P.R. China and has also been applied in previous studies (e.g., He et al., 2014; Qian et al., 2018) to represent the state of oxygen deficits, similar to the apparent oxygen utilization (AOU). For example, in the first paragraph of section 2.3.2 in He et al. (2014), they mentioned "The DO saturation (DO%) was calculated from the field-measured DO concentration divided by DO concentration at equilibrium with the atmosphere …"; in the second paragraph of section 3. 2 in Qian et al. (2018), they mentioned "In the winter, DO values were >200 mmol kg$^{-1}$ (>85% saturated) in the lower estuary throughout Transect C. In the spring, surface water DO values increased to >300 mmol kg$^{-1}$ and the DO saturation state reached >120% by May, …".

As suggested, we will provide further explanations to this metric (DOs) in our manuscript.

(3) **Section 3.1 and Figure 2:** Why not think about show AOU in the analysis?

Response: As we mentioned above, the oxygen saturation state (DOs, %) is also an indicator frequently used to reflect the state of oxygen deficits. In fact, it has a close relation with the AOU: AOU = (1-DOs)*DOsat (please note that DOsat represents the oxygen saturation concentration). According to the reviewer's suggestion, we have added AOU in Figure 2 (please see the revised figure below) and used for analysis as well in our manuscript.

[Figure]

Figure r1. Spatial means and standard deviations of DO concentrations, DO saturation (DOs), apparent oxygen utilization (AOU), salinity (S), and temperature (T) in the surface and bottom waters of the PRE in (a) spring (March-May) and winter (December-February), (b) summer (June-August), and (c) autumn (September-November) during 1976-2014. Note that the red dots in the first row of the figure represent the lowest DO values measured in each time period.

(4) **Line 148:** "The existence of hypoxic events in periods other than summer". The statement was kind of misleading. It seems it only happened in September 2006, not something unified exist. Please emphasize and rewrite.

Response: As suggested, this sentence will be rewritten as "*This reveals the existence of potential hypoxic events in periods other than summer*".

(5) **Line 165:** "the observed areas" and the following area number reported. The software used for the plots, and interpolation method to generate the low oxygen area should be well reported in the method section.

Response: The software we used for plotting includes MATLAB and EXCEL. As for the estimation on the low-oxygen areal extents in the PRE, our processing procedure is as follows: firstly, we divided the sea area of the PRE into a number of grid cells with a resolution of 0.01°, and then used the scattered-data-interpolation method (namely, the 'scatteredInterpolant' function) provided by MATLAB to interpolate the observational oxygen data onto the grid cells; secondly, we calculated the total areas for all the grid cells being hypoxic (with DO < 2 mg/L) to estimate the hypoxic areas. Same procedures were applied to compute the areal extents for oxygen deficiency and low oxygen by using a DO threshold of 3 and 4 mg/L, respectively.

As suggested, we will provide the above information on the software used for plotting and the interpolation method to estimate the low-oxygen areas in our manuscript.

(6) **Line 175:** I am confused about the statement "of which 1997, 2006 and 2013 have been shown earlier and will not be repeated here" please rewrite and clarify.

Response: As suggested, this sentence will be rewritten as "*note that the distributions in 1997, 2006 and 2013 have been shown in Figures 3-4 and will not be repeated here*".

(7) **Line 180:** This is a very interesting phenomenon reported. Figure 11a should be cited here also.

Response: The phenomenon reported in line 180 is about the low-oxygen levels in the surface waters observed in July 2005 and August 2013. We guess that what the reviewer actually wants to suggest is to cite Figure 2b1, not Figure 11a (please note that this subplot shows the wastewater discharge, not the oxygen data). Accordingly, we will add a citation of Figure 2b1 here.

(8) **Line 266-269:** The explanations of Figure 7b1 and 7b2. This was also because of the convergence induced by cyclonic vortices in the coastal transition zone (CTZ). Please add

some discussions.

Response: Thank you for the comment. We will add some discussions on the convergence induced by cyclonic vortices in the coastal transition zone (CTZ).

(9) **Section 4. Discussion.** I would expect to see a discussion on comparing long-term trend hypoxia variability with other systems, including both Chesapeake Bay and Gulf of Mexico. Please add section in this part.

Response: We will revise it as suggested. Please see our response to the General Comment 3 for details.

(10) **Table 2:** The definition of Pearson correlation coefficient should be explained in the method section. The correlation with NH4, NO3, PO4, is it with the nutrient concentration or with the loading? The details like this should be provided.

Response: The correlation analysis was performed on the oxygen and nutrient concentrations. As suggested, we will explain the definition of Pearson correlation coefficient in the method section of our manuscript, and will make it clear that the correlation analysis was carried out on the oxygen and nutrient concentrations in Table 2 of our manuscript.

(11) **Figure 10:** why the comparison was done between July 1999 and Sep 2006 in this figure? different year and different season. The pure bottom dissolved oxygen concentration should also be placed along with other variables

Response: Here we intended to use a combination of physical factors (including salinity and vertical density gradient) and biochemical-related factors (including the oxygen saturation states, SSC, and chlorophyll concentrations) to illustrate the differences between the summer and early autumn in terms of the mechanisms and key factors controlling the occurrence of low-oxygen conditions in the PRE. The reason for comparing these seasonal data in different years is simply due to the data limitations (a lack of long-term chlorophyll data). Currently, we only have chlorophyll data in July 1999 and September 2006 on hand (as listed in Table 1 of our manuscript). In spite of the data limitations, Figure 10 did show the distinct patterns of physical environments, SSC and chlorophyll distributions and their plausible effects on the low-oxygen conditions for different seasons (i.e. the summer and early autumn).

In regard to the bottom oxygen concentrations in July 1999 and September 2006, please note that they have already been presented in Figures 6d1 and 8b2. Therefore, in order to avoid the repeated display of the same information, the DO distributions were not added into Figure 10.

(12) **Figure 11:** Please provide a nutrient loading figure along with other variables.

Response: Thank you for the comment. We agree that it will be helpful to further explore the link between the long-term expansion of low-oxygen conditions and the environment changes in the Pearl River region by incorporating the nutrient loading (if applicable) along with river discharge

and sediment load into Figure 11. Unfortunately, the long-term nutrient loading data are not available. Instead, we provided the nutrient concentrations near the eastern four river outlets along with the wastewater discharge to reflect the pressure of anthropogenic pollutant inputs. Please see the revised figure (Figure r2) below (please note that the estimated areas of low-oxygen conditions during 1985-2017 were also added into the figure as suggested by the reviewer #1). There was an increasing trend in the nutrient concentrations along with the wastewater discharge. Although there existed data gaps in certain years, it is still clear that the nutrient concentrations after 2000 are higher than those before. This finding is also supported by Li et al. (2020). We have cited their findings regarding the changes in nutrients in our manuscript (lines 361-364).

[revised manuscript text omitted]

---

## Referee Report (RR1)

I appreciated the author team answered the question very responsible, including both my review comments and reviewer 1's comments. The author has published several papers on this topic on BG, most of them are model based. Although as a pure data analysis work, the paper is kind of old style for hypoxia science overall (I guess that is why the review request of the paper was turned down by many other researchers), it still deserved being published on BG as the first piece of work compiling long-term data for PRE hypoxia. However, I think the paper has two major technique issues and should be solved thoroughly before final acceptance on BG.

(1) Figure 1: I found that figure was used by the author team multiple times in different journals. For example:

Liang, B., Hu, J. T., Li, S. Y., Ye, Y. X., Liu, D. H., & Huang, J. (2020). Carbon system simulation in the Pearl River Estuary, China: mass fluxes and transformations. Journal of Geophysical Research: Biogeosciences, 125, e2019JG005012. https://doi.org/10.1029/2019JG005012

Figure 1 was the same as this one. That should be replot.

(2) The data conformity and availability issue. I do not think complicated method, e.g., numerical model and machine learning system, are only solutions. Although all extrapolation of oxygen data into unobserved stations will introduce uncertainties, there has been many other advanced statistical methods to solve the extrapolation problem specially

One example is:

Obenour DR, Scavia D, Rabalais NN, Turner RE, Michalak AM. Retrospective analysis of midsummer hypoxic area and volume in the northern Gulf of Mexico, 1985-2011. Environmental Science & Technology. 2013 Sep;47(17):9808-9815. DOI: 10.1021/es400983g. PMID: 23895102; PMCID: PMC3823027.

I believed because the observational data availability and data quality issue in Chinese coastal community. Collecting data and processing them are all really a lot of work for one piece of publication. I was OK with reviewer's response. But the author team should really make the data available on site

The data availability statement "The in-situ observation in July 1999 and 2013-2014 will be available at a public data storage, while ⋯" is not acceptable for modern top research journal these days. There should be an ftp website with last access date and checked by both reviewers.

I personal felt that the data transparency issue impeded the Chinese community promote the coastal science. A real opening data will be helpful for researchers to

work together to promote the estuary-coastal ocean science to a world leading level. The Chinese community do not really lack number of papers these years, isn't it? Did hypoxia community in other parts of world learn anything from it?

(3) The method part reads tedious in the new version and draw out the attention for the science itself. I noticed reviewer 1 challenged the data quality issue. I suggest move Line 111 to Line 125 to the supplementary.

I think all other questions are answered very well.

---

## Referee Report (RR2)

The data availability is an active link. Figure 1 has been updated. I think it was OK to publish on BG now and will be a useful piece of work for the Pearl River Hypoxia Study.

---

## Author Response (AR2)

For the second revision of our manuscript, we would like to thank the editor and two referees again for the positive comments and suggestions. Detailed response to all comments are given below (responses are shown in blue and relevant changes are marked in red in the revised manuscript).

**Anonymous Referee #1**

**Comments**

The points I raised are basically addressed. I have two concerns still.

1. table 2 mentioned the statistics between DO and some environmental parameters. I am curious that for samples in 2006 sep and 2001 sep, why DO showed contradictory relation with bottom S? namely 0.6637* in sep 2006 whereas -0.2953 in sep 2001. similar conflicting things also can be found for T and NO3.

Response: Thank you for providing these comments. We have noticed that the correlations between DO and other water quality variables (including salinity, temperature, and $NO_3$) in the bottom waters in September 2006 were opposite to those in September 2001. Such contradictory relation suggests that the key factors controlling low-oxygen conditions in the Pearl River Estuary (PRE) were likely different between these two early-autumn periods (also with different spatial patterns of DO and low-oxygen conditions; please see Figure 8 in our manuscript). Specifically, in September 2006, the bottom DO showed a significantly positive correlation with salinity (Table 2 in our manuscript), indicating the significant impact of river discharge. As affected by the river discharge (with higher temperature and nutrient concentrations but low DO levels when compared to the seawater end-members), the bottom DO therefore had a significantly negative correlation with temperature and a significantly positive correlation with $NO_3$. With respect to September 2001, no significant correlation was found between the bottom DO and environmental parameters. However, it is noted that the bottom DO showed a relatively good correlation with $NH_4$ among the environmental factors, which implies that the low-oxygen events on the eastern side of the PRE (Figure 8b1) were mainly resulted from the sewage effluents discharged from the adjacent coastal cities (i.e. Shenzhen and Hong Kong). This could be further supported by the significant correlation between the surface DO and $NH_4$ (with the correlation coefficient $r$ reaching -0.5686, $p < 0.05$; please see Table 2 in our manuscript). Collectively, the DO content was relatively low on the eastern side of the PRE, where salinity was relatively high (please see Figure r1 below). Therefore, the bottom DO exhibited a negative correlation with salinity (although not significant) overall in September 2001, which was opposite to that in September 2006.

[Figure]

Figure r1. Salinity distributions in the (a) surface and (b) bottom waters of the PRE for September of 2001 (left panels) and 2006 (right panels).

As we have mentioned in our manuscript, the differences between September of 2001 and 2006 indicate that in the periods of early autumn, there was considerable interannual variability in the spatial extents and intensity of low-oxygen conditions (and also the underlying mechanism as discussed above). At current stage, there is still a lack of in-depth investigation on the formation processes, interannual variations and driving factors of low-oxygen conditions in the early autumn of the PRE. We urge that more attention should be paid to this issue in future studies.

Based on the reviewer's comment, we have provided more discussions on the differences between these two early-autumn periods (please see lines 367-371 in our revised manuscript).

2. I suggest authors integrate and upload their 1976-2017 raw DO and environmental data that mentioned in current ms to a public repositories like figshare or any other free-downloaded and open source, in order to make the data accessible to public. Current data availability statement is strict to a few years sources (1999, 2013, 2014) while the strength of the paper, especially the title and how authors claims, clearly is based on a 1976-2017 time series data set.

Response: As suggested, we have made the observational data during 1976-2014 (Datasets 1-4 as listed in Table 1) available at https://doi.org/10.5281/zenodo.5195759 (latest access on August 15, 2021). Please note that the oxygen data in July of 2014-2017 (Dataset 5) derived from literatures can be downloaded directly via the links provided in the corresponding literatures.

**Anonymous Referee #2**

**Comments**

I appreciated the author team answered the question very responsible, including both my review comments and reviewer 1's comments. The author has published several papers on this topic on BG, most of them are model based. Although as a pure data analysis work, the paper is kind of old style for hypoxia science overall (I guess that is why the review request of the paper was turned down by many other researchers), it still deserved being published on BG as the first piece of work compiling long-term data for PRE hypoxia. However, I think the paper has two major technique issues and should be solved thoroughly before final acceptance on BG.

Response: We wish to thank the reviewer again for the positive comments and recognition of this work. We have revised the manuscript as suggested.

(1) Figure 1: I found that figure was used by the author team multiple times in different journals. For example:

Liang, B., Hu, J. T., Li, S. Y., Ye, Y. X., Liu, D. H., & Huang, J. (2020). Carbon system simulation in the Pearl River Estuary, China: mass fluxes and transformations. Journal of Geophysical Research: Biogeosciences, 125, e2019JG005012. https://doi.org/10.1029/2019JG005012

Figure 1 was the same as this one. That should be replot.

Response: Figure 1 did use the same base map as the one in Liang et al. (2020), which also focused on the Pearl River Estuary (PRE), but they are not exactly identical (e.g., the study area being shown). However, we have to admit that these figures still have a certain degree of similarity (especially the color map in use) and we fully understand the concerns from the reviewer and the editor. Therefore, in order to avoid the duplication, we have replotted Figure 1, using a different color map for the bathymetry with different contour levels. Please see the revised figure below.

[Figure]

Figure r1. Map of the Pearl River estuary (PRE) and adjacent coastal waters. Note that the purple dots denote cities in the Guangdong-Hong Kong-Macao Greater Bay Area, and the blue stars indicate the locations of eight outlets of the Pearl River freshwater discharged into the PRE; Humen, Jiaomen, Hongqili, and Hengmen are typically called the eastern four river outlets, while the others are called the western four river outlets.

(2) The data conformity and availability issue. I do not think complicated method, e.g., numerical model and machine learning system, are only solutions. Although all extrapolation of oxygen data into unobserved stations will introduce uncertainties, there has been many other advanced statistical methods to solve the extrapolation problem specially

One example is:

Obenour DR, Scavia D, Rabalais NN, Turner RE, Michalak AM. Retrospective analysis of midsummer hypoxic area and volume in the northern Gulf of Mexico, 1985-2011. Environmental Science & Technology. 2013 Sep;47(17):9808-9815. DOI: 10.1021/es400983g. PMID: 23895102; PMCID: PMC3823027.

I believed because the observational data availability and data quality issue in Chinese coastal community. Collecting data and processing them are all really a lot of work for one piece of publication. I was OK with reviewer's response. But the author team should really make the data available on site.

The data availability statement "The in-situ observation in July 1999 and 2013-2014 will be

available at a public data storage, while …" is not acceptable for modern top research journal these days. There should be an ftp website with last access date and checked by both reviewers. I personal felt that the data transparency issue impeded the Chinese community promote the coastal science. A real opening data will be helpful for researchers to work together to promote the estuary-coastal ocean science to a world leading level. The Chinese community do not really lack number of papers these years, isn't it? Did hypoxia community in other parts of world learn anything from it?

Response: We totally agree with the reviewer's comments on the importance of data sharing and the issue related to data transparency. In fact, it took us several years and tremendous efforts to collect the observational data used in this study. This is not a technical issue at all, but it is a very important and basic step that enables us to promote the research progress on the long-term oxygen changes in the PRE. It is also our hope to have more data available and accessible so that we could fill the data gaps to better quantify the decadal changes in low-oxygen conditions in the PRE and clarify the key mechanisms controlling their expansions over recent years.

As suggested, we have made the observational data during 1976-2014 (datasets 1-4 as listed in Table 1) available at https://doi.org/10.5281/zenodo.5195759 (latest access on August 15, 2021). Please note that the oxygen data in July of 2014-2017 (dataset 5) derived from literatures can be downloaded directly via the links provided in the corresponding literatures.

(3) The method part reads tedious in the new version and draw out the attention for the science itself. I noticed reviewer 1 challenged the data quality issue. I suggest move Line 111 to Line 125 to the supplementary.

Response: As suggested, we have moved lines 111-125 to the supplementary materials of our revised manuscript.

I think all other questions are answered very well.

Response: Thank you for your support.